# Context-aware Dynamic Pruning for Speech Foundation Models

**Masao Someki, Yifan Peng, Siddhant Arora, Shinji Watanabe**
Language Technologies Institute
Carnegie Mellon University
Pittsburgh, PA 15213, USA
{msomeki,yifanpen,siddhana}@andrew.cmu.edu, shinjiw@ieee.org,

**Markus Müller, Thanasis Mouchtaris, Grant Strimel, Jing Liu**
Neural Efficiency Science
Amazon Artificial General Intelligence
Pittsburgh, PA 15203, USA
{mumarkus,mouchta,gsstrime,jlmk}@amazon.com

## Abstract

Foundation models, such as large language models, have achieved remarkable success in natural language processing and are evolving into models capable of handling multiple modalities. Listening ability, in particular, is crucial for many applications, leading to research on building speech foundation models. However, the high computational cost of these large models presents a significant challenge for real-world applications. Although substantial efforts have been made to reduce computational costs, such as through pruning techniques, the majority of these approaches are applied primarily during the training phase for specific downstream tasks. In this study, we hypothesize that optimal pruned networks may vary based on contextual factors such as speaker characteristics, languages, and tasks. To address this, we propose a dynamic pruning technique that adapts to these contexts during inference without altering the underlying model. We demonstrated that we could successfully reduce inference time by approximately 30% while maintaining accuracy in multilingual/multi-task scenarios. We also found that the obtained pruned structure offers meaningful interpretations based on the context, e.g., task-related information emerging as the dominant factor for efficient pruning.

## 1 Introduction

In recent years, foundation models have achieved remarkable success across various tasks in natural language processing (OpenAI, 2022; 2023; Google, 2023; Meta, 2024; Anthropic, 2024; Amazon, 2024; DeepSeek-AI, 2025). These Large Language Models (LLMs) have been particularly effective as multi-modal systems, incorporating modalities such as images and videos (Google, 2024; Anthropic, 2024; Amazon, 2024). The integration of voice as a modality for communication between humans and LLMs has also gained traction, leading to applications that facilitate interactive conversations with LLMs (OpenAI, 2024; Défossez et al., 2024). Many studies have explored features to integrate the hearing ability into LLMs, employing methods such as connecting massive audio encoders to LLMs (Changli et al., 2024; Yuan et al., 2024; Chu et al., 2023; HU et al., 2024; Défossez et al., 2024) and utilizing large speech-to-text foundation models with powerful multilingual and multi-task capabilities (Radford et al., 2023; Peng et al., 2023d; Puvvada et al., 2024).

However, this broad support necessitates the training of large-scale models with billions of parameters, introducing new challenges such as increased inference costs. In speech processing, where input sequences tend to be longer than those in language processing, computationally intensive models can significantly prolong inference times. To address these challenges, various methodologies have been proposed, including bifocal networks (Macoskey et al., 2021a), dual-attention architectures (Sahai et al., 2023), amortized networks (Macoskey et al., 2021b; Xie et al., 2022; Strimel

et al., 2023), pruning (Fu et al., 2022; Lai et al., 2021; Peng et al., 2023a; Wang et al., 2023; Ding et al., 2024), quantization (Nguyen et al., 2020; Ding et al., 2024), distillation (Liu et al., 2021; Chang et al., 2022; 2024; Gandhi et al., 2023), and combinations of those methods (Peng et al., 2023c). However, these approaches primarily focus on reducing the model size during training.

While these models handle various tasks, it raises a fundamental question: *is a single model optimal structured for all tasks and languages?* Different languages and tasks may require unique pruning strategies for effective processing. For example, Chen et al. (2022) highlights that different layers contribute differently depending on the task, indicating that the optimal model structure might vary across tasks. They also showed excellent performance in ASR using the WavLM encoder with only a single linear layer as the decoder, which raises questions about the need for a large-scale decoder in ASR systems. Conversely, Peng et al. (2024a) highlight the importance of the decoder network in ST, suggesting that the decoder may play a more critical role in ST compared to ASR. Therefore, we hypothesize that there might be an optimal model structure depending on each task and language combination so that each subnetwork has the potential to perform with comparable accuracy with less inference complexity.

Given this hypothesis, we propose a method for *dynamically* pruning a pre-trained foundation model based on the context information, including speech features, language, and task characteristics, enabling the construction of an optimal model architecture tailored to the contextual requirements *during inference*. Specifically, we train a model that computes module-level masks for each layer in the encoder and decoder networks based on the provided context while simultaneously fine-tuning the foundation model. The predicted mask is utilized to determine which modules to activate or skip while maintaining accuracy. By analyzing the pruned network, we offer an interpretation of the importance of the optimal subnetwork within the given contexts.

This paper makes the following key contributions:

1. We propose to apply a novel context-aware pruning technique to each module in a speech foundation model dynamically within multilingual and multi-task scenarios.

2. We were able to reduce inference time by 34.3% without degrading the BLEU scores for the ST task and 28.6% with only 2.8% WER degradation on the ASR task.

3. We conducted a detailed comparative analysis and found that the obtained pruned structure offers meaningful interpretations based on the context, e.g., task-related information emerging as the dominant factor for efficient pruning.

## 2 RELATED WORK

Pruning techniques are mainly classified into unstructured and structured approaches. The former is a technique for deleting individual weights in a network (LeCun et al., 1989; Hassibi et al., 1993; Han et al., 2016b); however, it has a problem of low compatibility with hardware accelerators (Han et al., 2016a; Liu et al., 2024). This method is further investigated in Appendix D. On the other hand, structured pruning has a more direct benefit in reducing the complexity, which performs pruning on a layer or module basis, including filters/layers in CNNs (Wen et al., 2016; Li et al., 2017; Alvarez & Salzmann, 2016; Han et al., 2017) or layer-wise pruning in models (Fan et al., 2020; Lee et al., 2021; Chen & Zhao, 2019). Thus, our paper employs structured pruning.

Methods for determining pruning targets include gradient-based techniques (Guo et al., 2016; He et al., 2020; Fu et al., 2022; Wen et al., 2016) as well as magnitude-based approaches for modules (Li et al., 2017; 2022). However, these methods typically use a fixed architecture during inference. Addressing this limitation, recent research has focused on implementing efficient inference by dynamically adjusting the computational load during the inference process (Bengio et al., 2016; Jernite et al., 2017; Bolukbasi et al., 2017; Graves, 2016). Notably, in the speech domain, numerous studies have explored streaming models to achieve dynamic model structures aimed at speedup (Macoskey et al., 2021a;b; Strimel et al., 2023; Xie et al., 2022; Xu et al., 2023). In this context, Peng et al. (2023b); Bittar et al. (2024) extended this concept to large-scale Transformer-based models, exploring dynamic layer-wise structural changes to enhance efficiency. In our study, we extend the work of Peng et al. (2023b) by utilizing the model structure of a speech foundation model to address multilingual and multi-task scenarios. This extension explores how a large-scale speech foundation model adapts its structure based on context and input condition (Lu et al., 2024), providing insights

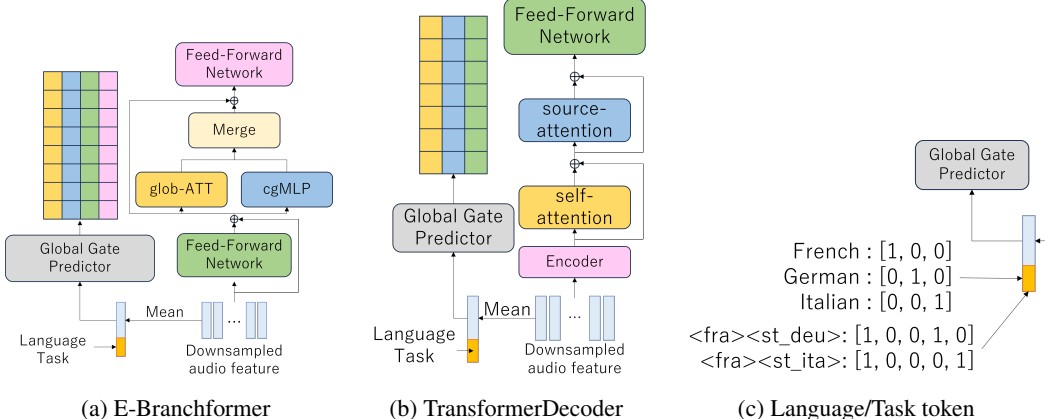

(a) E-Branchformer     (b) TransformerDecoder     (c) Language/Task token

Figure 1: The sparse E-Branchformer and Transformer architectures in the experiment, and the method for embedding the language/task information. The audio information and the language/task information are concatenated, and the Gate Predictor calculates the gate probability for each module in each layer.

into more efficient and context-aware (Chang et al., 2021; Sathyendra et al., 2022) speech processing systems.

## 3 METHODS

### 3.1 OPEN WHISPER-STYLE SPEECH MODELS

In this study, we utilized Open Whisper-Style Speech Models (OWSM) (Peng et al., 2023d) as the foundation for our speech model experiments. OWSM is an open-source reproduction of OpenAI's Whisper model (Radford et al., 2023). Among the available versions, we selected OWSM-v3.1 (Peng et al., 2024b) as the speech foundation model for our experiments. The key rationale for selecting OWSM models lies in its fully open training data, processes, and configuration compared to Whisper. We ensure that the data used for validation in our experiments is not part of the pre-training corpus. This transparency is critical. If validation data was included in the pre-training data, it could inflate post-pruning accuracy, hindering a proper evaluation of the pruned network's performance.

Additionally, the E-Branchformer architecture integrated into OWSM-v3.1 offers a more flexible and generalized structure compared to the conformer, due to its parallel design. The model employs a dual-branch structure: one branch extracts global context using a self-attention-based module (glob-ATT), while the other captures local context using a convolution-based module (cgMLP) Sakuma et al. (2022). These branches are merged through a convolution-based merging layer, and are enclosed between two feed-forward networks (FFN1 and FFN2). Through the elimination of particular modules in the E-Branchformer layers, we can create a model resembling a conformer. This will enable us to thoroughly evaluate the Transformer's efficiency and effectiveness as an architectural design.

### 3.2 MODULE-LEVEL PRUNING

We implement module-level pruning in our study, targeting essential components within foundation models like self-attention (ATT) and feed-forward networks (FFN). For example, in the Transformer architecture, we prune the self-ATT, source-attention (src-ATT), and FFN as modules. In the case of the E-Branchformer, pruning modules include the FFN1, glob-ATT, cgMLP, and the FFN2.

The motivation for adopting module-wise pruning is our assumption that the model's architecture should remain flexible and adaptable during inference. Pruning techniques that operate at a finer granularity, such as kernel pruning or layer pruning, which removes individual kernels or layers

from convolution components, would disrupt the model's structural integrity and limit its ability to dynamically adapt to different audio inputs. While layer-wise pruning aligns with our goal of simplifying the model, it oversimplifies the pruned model and prevents us from observing the importance of individual modules. For these reasons, we decided to employ module-wise pruning techniques to balance between structural flexibility and model interpretability. Further considerations on layer-skip approaches have been included in the Appendix C for reference.

### 3.3 PRUNING

Given the dynamic nature of the input speech, it is necessary to generate a mask for each module based on audio input, language, and task information. To achieve this, we employ a neural network model to estimate a binary mask that determines whether to use a module. We frame the pruning problem as an L0 regularization task Louizos et al. (2018), optimizing the expected value of the binary mask to achieve the desired sparsity. While Louizos et al. (2018) uses a Sigmoid-based approach, we follow Peng et al. (2023b) for implementation efficiency and treat the mask estimation as a two-class classification problem using Gumbel-Softmax Jang et al. (2017) for implementation efficiency.

In these works, pruning masks were learned using the sigmoid function or Gumbel-Softmax. However, the masks used during training were continuous values between 0 and 1, rather than strict binary values. As a result, modules that should have been completely skipped during inference were still partially utilized during training. During fine-tuning the OWSM model, we observed that even with a very low temperature for the softmax operation for probabilities, the gate probabilities often remained in the range between 0.4 and 0.6. This led to a discrepancy where the output of a module was scaled by a factor of 0.4 during training, while the same module was entirely skipped during inference because the probability fell below the threshold value, such as 0.5. To address this issue, we employed the Straight-through Gumbel-Softmax Estimator (SGSE) Jang et al. (2017) to ensure that the output of the gate predictor was strictly binary. With SGSE, the forward pass computations are performed using binary values, while the backward pass estimates gradients with continuous values, allowing the model to be trained effectively. The detailed formulation is provided in Appendix A.

In our work, inspired by Peng et al. (2023b) and Wang et al. (2020), we define the sparsity loss function, $\mathcal{L}_{\text{sparsity}}$, as follows:

$$\mathcal{L}_{\text{sparsity}} = \alpha\{|g - s_{\text{target}}| + (g - s_{\text{target}})^2\}, \tag{1}$$

where $g$ is the average of the gate probabilities for all modules, $\alpha$ refers to the weight for $\mathcal{L}_{\text{sparsity}}$, and $s_{\text{target}}$ is the desired sparsity ratio for the model. Since we use Gumbel-Softmax to binarize all gate probabilities, $g$ represents the proportion of modules that are activated in the entire model, i.e., the model's sparsity ratio. For a detailed derivation of the loss function, please refer to the Appendix E.

Additionally, when both the Encoder and Decoder are pruned, we calculate the $\mathcal{L}_{\text{sparsity}}$ based on the gate probabilities from both components to bring the overall model sparsity closer to $s_{\text{target}}$. Let $g_{\text{enc}}$ be the gate probability for any module in the encoder, and $g_{\text{dec}}$ be the gate probability for any module in the decoder. Then, the $L_{\text{sparsity}}$ is calculated for three scenarios, from top to bottom: first, when only the encoder is pruned; second, when only the decoder is pruned; and third, when both the encoder and decoder are pruned simultaneously.

$$\mathcal{L}_{\text{sparsity}} = \begin{cases} \alpha \left|\mathbb{E}[g_{\text{enc}}] - s_{\text{target}}\right| + \alpha \left(\mathbb{E}[g_{\text{enc}}] - s_{\text{target}}\right)^2 & \text{encoder only} \\ \alpha \left|\mathbb{E}[g_{\text{dec}}] - s_{\text{target}}\right| + \alpha \left(\mathbb{E}[g_{\text{dec}}] - s_{\text{target}}\right)^2 & \text{decoder only} \\ \frac{\alpha}{2} \left|\mathbb{E}[g_{\text{enc}}] + \mathbb{E}[g_{\text{dec}}] - 2s_{\text{target}}\right| + \frac{\alpha}{4} \left(\mathbb{E}[g_{\text{enc}}] + \mathbb{E}[g_{\text{dec}}] - 2s_{\text{target}}\right)^2 & \text{jointly} \end{cases}$$

As noted by Wang et al. (2020), $s_{\text{target}}$ is gradually increased during training. Therefore, let the loss for the downstream task be $\mathcal{L}_{\text{owsm}}$, the overall loss function that we aim to minimize is:

$$\mathcal{L} = \mathcal{L}_{\text{owsm}} + \mathcal{L}_{\text{sparsity}}. \tag{2}$$

### 3.4 CONTEXT-AWARE GATE PREDICTOR

As shown in Fig. 1, the gate probability is calculated using Gate Predictors. In Peng et al. (2023b), two types of Gate Predictors are proposed: GlobalGP and LocalGP. GlobalGP calculates the gate

probability based on the encoder's input, which is also fed into the first layer of the encoder. In contrast, LocalGP provides a Gate Predictor for each layer, computing the probability based on the input to that specific layer. While both methods have shown promising results, we opted for GlobalGP due to its implementation simplicity. The detailed process of calculating gate probability is in Appendix G.

To handle multiple languages and tasks simultaneously, we created vectors representing the language and task, combined them with the speech features, and used them as input to the Gate Predictors. These vectors are combinations of one-hot vectors representing the language and task. For example, if there are two languages, French and German, and tasks including speech recognition and translation between them, the language conditions are $[0, 1]$ for French and $[1, 0]$ for German, and the task conditions are $[0, 0, 1]$ for speech recognition, $[0, 1, 0]$ for French to German translation, and $[1, 0, 0]$ for German to French translation. Combining these, tasks such as French speech recognition can be expressed as $[0, 1, 0, 0, 1]$, and similarly $[0, 1, 0, 1, 0]$ means translating French to German.

## 4 EXPERIMENTS

### 4.1 EXPERIMENTAL SETUP

**Dataset**  This study employs the Europarl-ST (Iranzo-Sánchez et al., 2020) dataset to evaluate model performance across multiple languages. The corpus was compiled from debates held in the European Parliament between 2008 and 2012. We utilized version 1.1 of the dataset, which comprises speech data in nine languages. For our experiments, we selected German, French, and Italian, which consist of approximately 20 hours of speech data. As the europarl-ST dataset is not part of the OWSM training data, we deemed it suitable for evaluating the model under multi-lingual and multi-task settings.

**Task**  We fine-tuned the OWSM model with a pruning objective across ASR and ST tasks. The experiments were designed to compare two pruning strategies: one in which the model was trained on ASR and ST tasks independently, and another where both tasks were integrated during training. Additionally, we investigated the effects of sparsifying the model in three configurations: sparsifying only the Encoder, only the Decoder, and both simultaneously.

**Evaluation**  In this experiment, we evaluated the ASR task using Word Error Rate (WER) and the ST task using BLEU scores. For each language, we prepared models with sparsity ratio of 10%, 30%, 50%, 70%, and 90%, and assessed their performance. Additionally, we used a baseline model that was fine-tuned with all modules retained for comparison. Note that the sparsity level refers to the ratio of activated modules to the total number of modules, not the number of parameters in the model. In all experiments, we performed auto-regressive decoding with a beam size of 5.

The model sparsity observed in this experiment is visualized using heat maps, where each cell represents the gate probability for all modules across all layers. During model validation, the activation frequency of each gate is accumlated, and the average is computed to derive the expected activation probability per module.

### 4.2 RESULTS

#### 4.2.1 MULTI-LINGUAL ASR

Figure 2 shows the WER for German. In Figures 3 and 4, we present the visualization of $\mathbb{E}[g_{\text{enc}}]$ and $\mathbb{E}[g_{\text{dec}}]$ for each module when the encoder and decoder were pruned separately. Figure 5 illustrates the $\mathbb{E}[g_{\text{enc}}]$ and $\mathbb{E}[g_{\text{dec}}]$ when encoder and decoder were jointly pruned. We analyzed the $g$ across varied $s_{\text{target}}$ and languages, and found no significant differences. Therefore, we focus on the German ASR results here. For complete heatmaps and a WER table, refer to Appendix H. We considered the possibility that the differences in the amount of data used for pre-training the OWSM model across languages might affect our results and conducted additional training accordingly. The results are provided in Appendix B.

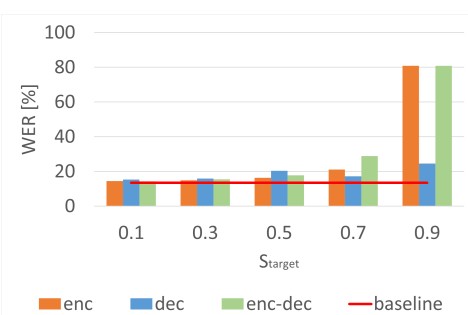

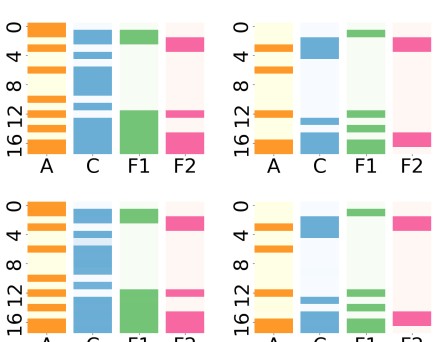

Figure 2: Comparison of sparsity ratio and WER. The label *enc* indicates pruning of the encoder only, *dec* indicates pruning of the decoder only, and *enc-dec* indicates simultaneous pruning of both. The WER is evaluated for German as the $s_{target}$ varies from 0.1 to 0.9. The baseline WER is 13.5. Decoder pruning retains WER even at high sparsity, while encoder pruning significantly degrades it.

Figure 3: Visualization of $\mathbb{E}[g_{enc}]$ when pruned separately with German and French ASR. The columns represents the $s_{target}$, with $s_{target}$ being 0.5 and 0.7 from left to right. The first row represents the result for German ASR, and the second row represents the French ASR. The y-axis within the heatmap represents the depth, where top is the first layer. The label *A* indicates glob-ATT, *C* indicates cgMLP, *F1* indicates FFN1, and *F2* indicates FFN2.

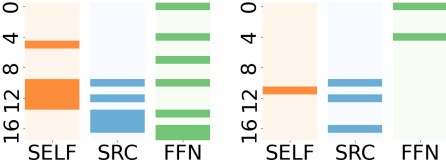

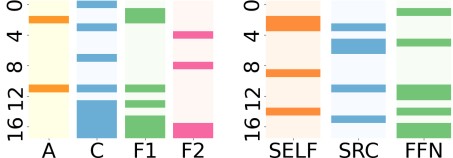

Figure 4: Visualization of $\mathbb{E}[g_{dec}]$ in German ASR, when decoder was pruned separately. The columns represent the $s_{target}$, with $s_{target}$ being 0.7 and 0.9 from left to right. The label *SELF* indicates self-ATT, *SRC* indicates src-ATT, and *FFN* indicates FFN. The other settings are consistent with those in Figure 3.

Figure 5: Visualization of $\mathbb{E}[g_{enc}]$ and $\mathbb{E}[g_{dec}]$ when they were pruned jointly. The $s_{targat}$ for this figure is 0.7. The left image corresponds to the encoder, and the right image corresponds to the decoder. The other settings are consistent with those in Figure 3 and Figure 4.

**Inference Performance**  We found that pruning the decoder side did not harm WER, even with high sparsity ratios, where pruning encoder modules greatly deteriorate the WER in high sparsity ratio. By analyzing the results alongside the module heatmap, we observed a decline in encoder accuracy at $s_{target} = 0.7$, specifically when cgMLP started to be pruned, underscoring the critical role of cgMLP in ASR tasks. In contrast, observing the decoder side with a $s_{target} = 0.9$, where a substantial number of FFNs have been pruned, we find that WER does not deteriorate as severely as in the ST case discussed in section 4.2.2. This finding supports our initial question that ASR may not require large-scale decoders to the same extent as ST.

**Sparse Encoder Analysis**  Referring to Figure. 3, the encoder's pruning strategy remained consistent across languages. Additionally, the highly polarized colors in Figure. 3, indicate that the gate probabilities are concentrated at extreme values, suggesting minimal variation in module selection based on speech characteristics. Interestingly, pruning approximately 50% of the encoder modules led to a more biased module selection, favoring cgMLP activations. This highlights the crucial role of local context captured by cgMLP, challenging the current architectural convention that equally balances both.

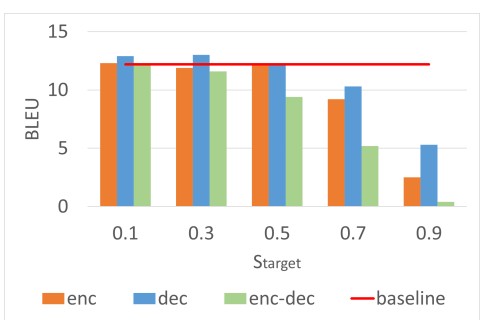

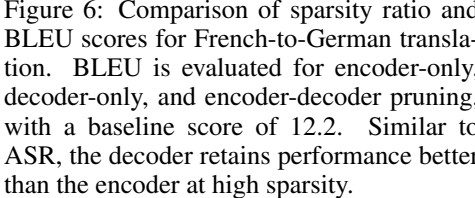

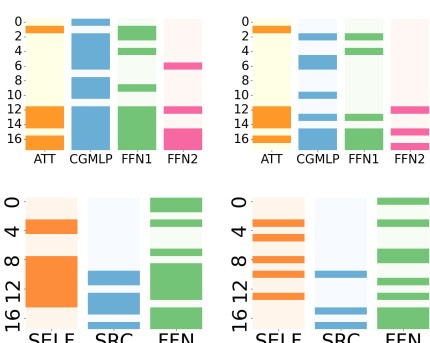

Figure 6: Comparison of sparsity ratio and BLEU scores for French-to-German translation. BLEU is evaluated for encoder-only, decoder-only, and encoder-decoder pruning, with a baseline score of 12.2. Similar to ASR, the decoder retains performance better than the encoder at high sparsity.

Figure 7: Visualization of $\mathbb{E}[g_{\text{enc}}]$ and $\mathbb{E}[g_{\text{dec}}]$ when they were pruned jointly. The columns represents the $s_{\text{target}}$, with $s_{\text{target}}$ being 0.5 and 0.7 from left to right. The first row represents the encoder, and the second row represents the decoder. The other settings are consistent with those in Figure 3.

**Sparse Decoder Analysis** In conditions where 90% of the modules were pruned, src-ATT was prioritized over self-ATT and FFN in Figure 4, indicating its essential role in inference. Given that src-ATT is the module responsible for incorporating audio information, this behavior is understandable. At a 70% sparsity ratio, we observed a typical flow in the TransformerDecoder, where the process moves from self-ATT to src-ATT, and then to FFN. The heatmap indicates that groups of modules can be computed in chunks, with several self-ATT modules processed together, followed by a block of src-ATT modules, and then a group of FFN modules. These findings suggest that incorporating chunk-wise computation could improve the efficiency of conventional decoder architectures.

**Combined Encoder-Decoder Sparsity Analysis** We found that several portions of the encoder exhibit a similar architecture to that of the Conformer. That is, the processing sequence progresses from the FFN to the glob-ATT, then to the cgMLP, and back to the FFN. This finding suggests that the Conformer architecture is effective in speech foundation models, particularly when the model size is constrained.

Compared to Figure 4 and Figure 5, we observed that when pruning is applied only to the decoder, src-attention layers in the early part of the decoder are often skipped. However, when both the encoder and decoder are pruned together, the earlier src-attention layers in the decoder become more active. This difference appears to stem from whether the encoder's full capacity is available. When the encoder is fully utilized, the decoder computes more self-attention and FFN layers before src-attention to incorporate additional contextual information from the output tokens. In contrast, when encoder capacity is limited, the decoder compensates by performing self-attention and FFN computations directly on the audio features to capture details that may have been missed by the encoder. Additionally, by analyzing the number of active modules in the decoder, we found differences in the number of FFN layers processed before src-attention, which further supports this interpretation. These findings are also supported by visualizations provided in the Appendix H.

### 4.2.2 MULTI-LINGUAL ST

Figure 6 presents the BLEU scores for the French-to-German translation tasks. Figure 7 visualize the $\mathbb{E}[g_{\text{enc}}]$ and $\mathbb{E}[g_{\text{dec}}]$ for each module when the encoder and decoder were pruned separately. We also analyzed the $g$ across varied $s_{\text{target}}$ and tasks, and found no significant differences. Therefore, we focus on the French-to-German ST results here. Refer to the Appendix H for a complete heatmaps and table of BLEU score. The analysis on combined encoder-decoder settings are also in the Appendix H.

**Inference Performance** Focusing on the encoder side in Figure 6, the BLEU scores remained relatively stable even with a $s_{\text{target}} = 0.5$. On the other hand, from Figure 6 and Figure 7, BLEU scores began to degrade when usage of cgMLP modules dropped when $s_{\text{target}}$ becomes 0.7. This highlights the importance of convolution-based models in ST, consistent with ASR tasks. A notable difference from ASR is that pruning the decoder also deteriorates model performance. We found that in the ST decoder, the FFN tends to be retained in computation. As the model removes the FFN, the BLEU score also degrades. This observation supports our initial hypothesis that the decoder plays a more critical role in ST compared to ASR.

**Sparse Encoder Analysis** In Figure 7, we observed an increased importance of cgMLP, similar to the findings in ASR. However, unlike ASR, the utilization of FFN2 decreased in the earlier layers of the ST task. Nevertheless, there were no significant differences in module selection within the encoder structure. This indicates that it may not be necessary to modify the encoder when designing models intended to handle both ASR and ST tasks simultaneously.

**Sparse Decoder Analysis** From $s_{\text{target}} = 0.5$ to $s_{\text{target}} = 0.7$, the number of activated src-ATT decreased significantly in Figure 7, with this reduction being larger than ASR. Additionally, it became apparent that self-ATT were computed over a broader range of depths compared to ASR. This suggests that while ASR places greater importance on audio information, ST requires self-ATT and FFN more than audio information for translation. This increased reliance can be attributed to the non-monotonic relationship between input audio and output text in ST, necessitating a greater use of self-ATT and FFN to capture complex dependencies.

In Figure 7, we observed that self-ATT layers are more frequently activated before src-ATT in ST compared to ASR. We hypothesize that this difference arises from the distinct priorities of each task. In ASR, the primary focus is on integrating audio features directly, as each computation of src-ATT increases the prominence of audio information as a weighted sum. In contrast, ST seems to place a higher importance on alignment text information through self-ATT before src-ATT to effectively map different languages. As a result, self-ATT layers are activated earlier to better contextualize before src-ATT, reflecting the task-specific demands of aligning cross-modal information. These findings underscore how the allocation of self-ATT and src-ATT computations is influenced by the differing requirements of ASR and ST.

### 4.2.3 PRUNING BY JOINT ASR AND ST

Figure 8 represents the WER for German ASR, and Figure 9 represents the BLEU score for French-to-German translation task. Figure 9 represents the BLEU score for French-to-German translation task. Same as the single-task settings, we also analyzed the $g$ across varied $s_{\text{target}}$ and tasks, but could not find differences in module selection across languages and tasks. Therefore, we focus on the French-to-German ST results here. Refer to the Appendix H for a complete heatmaps and tables for other tasks and languages. The analysis of the case where the encoder and decoder were pruned separately is included in Paragraph 4.2.3.

**Inference Performance** When ASR and ST tasks were trained simultaneously, a slight degradation in WER and BLEU was observed in Figures 8 and 9, particularly when pruning was applied to both the encoder and decoder. However, we found that when only the decoder was pruned, performance was better maintained compared to other settings. These findings suggest that, for large-scale speech foundation models trained on multiple tasks, focusing on decoder pruning is a more effective strategy for preserving accuracy across various tasks.

**Combined Encoder-Decoder Sparsity Analysis** In Figure 10, the processing order observed was FFN, followed by src-ATT, and then self-att, particularly when $s_{\text{target}}$ is 0.7. This order contradicts the typical processing sequence of a Transformer decoder and the observations made in 4.2.1. These results suggest that the conventional processing order of Transformer decoders may not be optimal for speech foundation models trained on multi-task data.

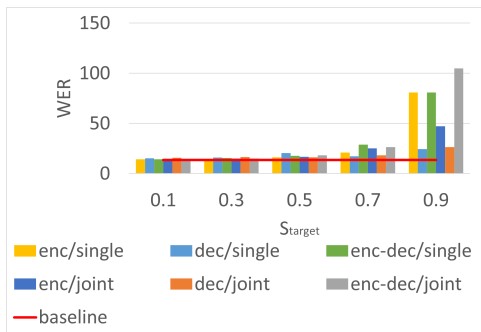

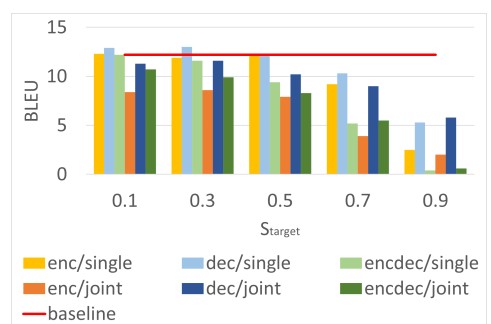

Figure 8: Comparison of $s_{\text{target}}$ and WER on German ASR. This figure compares results when using only ASR data versus using both ASR and ST data. *single* refers to the results obtained using only ASR data, and *joint* includes results from pruning that also incorporates ST data. The other settings are consistent with those in Figure 3.

Figure 9: Comparison of $s_{\text{target}}$ and BLEU on French-to-German ST. This figure compares results when using only ST data versus using both ASR and ST data. The labels are the same as Figure 8. Combined with Figure 8, this suggests that decoder pruning in multitask settings maintains competitive performance compared to task-specific pruning.

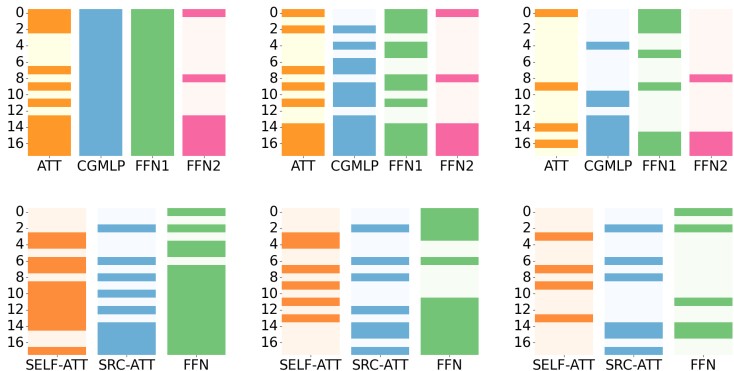

Figure 10: Visualization of $\mathbb{E}[g_{\text{enc}}]$ and $\mathbb{E}[g_{\text{dec}}]$ when they were pruned jointly. The columns represent the $s_{\text{target}}$, with $s_{\text{target}}$ being 0.3, 0.5, and 0.7 from left to right. The first row represents the encoder, and the second row represents the decoder. The other settings are consistent with those in Figure 3.

## 4.3 INFERENCE EFFICIENCY

We measured the actual inference time of the pruned model to analyze the effect of pruning on inference speed. We used one A40 GPU and 16 CPUs for each inference run. We employed vectorized beam search (Seki et al., 2019) for decoding, where the beam size is 5. Since previous experiments showed no variation in module selection across languages or tasks, we focused on one language and task, specifically German ASR for measuring inference time in each case. Table 1 presents the trends in metrics, inference time, and FLOPs as a function of module sparsity. It is important to note that we ignored the first run of inference, as it contains initialization processes that make it slower.

The results show that reducing the decoder modules by 50% improves latency while maintaining accuracy for both ASR and ST tasks. Specifically, we achieved a 34.3% reduction in inference time with no degradation in BLEU for ST, and a 28.6% reduction with only a 2.8% WER increase for ASR. Since OWSM uses auto-regressive inference with vectorized beam search, the decoder handles the majority of the computational load. This is reflected in the significant reduction in FLOPs observed during pruning, as the decoder processes each output token individually, treating the beam size as the batch size. In this analysis, we set the beam size to 5, meaning the encoder's

Table 1: Metrics and Elapsed time on inference for ASR (German) and ST (French-to-German) across varying $s_{\text{target}}$. The models were trained on task-specific training datasets. The table compares elapsed time, GFLOPs, and metrics (WER for ASR and BLEU for ST) at different $s_{\text{target}}$ for the enc, dec, and enc-dec. Baseline results are: ASR - 9.28 seconds and 13.5 WER, and 3781 GFLOPs; ST - 10.54 seconds, 12.2 BLEU, and 3409 GFLOPs. Each row represents a different sparsity target, showing the impact on inference time and output quality as the sparsity increases. *ET* refers to the elapsed time. We use `fvcore` library to estimate the GFLOPs.

| | ASR (German) | | | | | | | | |
| | Encoder | | | Decoder | | | EncDec | | |
| sparsity | ET | GFLOPs | WER | ET | GFLOPs | WER | ET | GFLOPs | WER |
|---|---|---|---|---|---|---|---|---|---|
| 10% | 9.07 | 3697 | 14.4 | 9.33 | 3268 | 15.3 | 10.39 | 2843 | 14.3 |
| 30% | 9.21 | 3690 | 14.8 | 7.24 | 2293 | 15.9 | 7.09 | 2249 | 15.4 |
| 50% | 8.92 | 3669 | 16.3 | 6.62 | 1713 | 20.4 | 5.99 | 1698 | 17.8 |
| 70% | 9.15 | 3633 | 21.0 | 4.79 | 1272 | 17.2 | 5.49 | 1139 | 28.8 |
| 90% | 8.52 | 3613 | 80.8 | 4.80 | 625 | 24.5 | 5.22 | 682 | 80.8 |

| | ST (French-to-German) | | | | | | | | |
| | Encoder | | | Decoder | | | EncDec | | |
| sparsity | ET | GFLOPs | BLEU | ET | GFLOPs | BLEU | ET | GFLOPs | BLEU |
|---|---|---|---|---|---|---|---|---|---|
| 10% | 10.42 | 3369 | 12.3 | 10.72 | 2718 | 12.9 | 10.58 | 2861 | 12.2 |
| 30% | 10.08 | 3357 | 11.9 | 8.27 | 2015 | 13.0 | 9.27 | 2093 | 11.6 |
| 50% | 10.27 | 3335 | 12.3 | 6.92 | 1521 | 12.2 | 7.09 | 2023 | 9.4 |
| 70% | 10.38 | 3311 | 9.2 | 5.38 | 1063 | 10.3 | 6.58 | 936 | 5.2 |
| 90% | 10.44 | 3298 | 2.5 | 4.27 | 504 | 5.3 | 4.81 | 549 | 0.4 |

batch size is 1, while the decoder's is 5. As a result, pruning the decoder not only reduces FLOPs but also has a more pronounced impact on inference speed compared to pruning the encoder.

# 5 CONCLUSION

In this work, we proposed a novel context-aware dynamic pruning method for speech foundation models that adapts pruning dynamically during inference. With the pruned model, we successfully accelerated the inference of speech foundation models, particularly without any degradation in the ST task. Through a detailed analysis of the model structures that emerge after pruning, we identified the efficiency of the Transformer decoder and Conformer, while also uncovering an interesting computational flow when the model was pruned in multi-task settings. Although this study focused on the speech domain, our approach can be readily extended to foundation models in other fields, such as NLP and computer vision.

# 6 ACKNOWLEDGEMENTS

This study was supported by the BRIDGE program of the Cabinet Office, Government of Japan. Also, we used the Bridges2 system at PSC and Delta system at NCSA through allocation CIS210014 from the Advanced Cyberinfrastructure Coordination Ecosystem: Services & Support (ACCESS) program, which is supported by National Science Foundation grants #2138259, #2138286, #2138307, #2137603, and #2138296.

## 7 REPRODUCIBILITY

You can download the OWSM-v3.1 we employed in this experiment from the huggingface hub [1]
All of our experiments are conducted with ESPnet (Watanabe et al., 2018). Based on the training
configuration of OWSM-v3.1, we added or modified the following configuration:

```
encoder: e_branchformer_token_condition
decoder: transformer_decoder_token_condition

tau_ini: 1
tau_end: 0.1
tau_cooldown_steps: 15000
sparsity_init: 0.0
sparsity_end: 0.3

optim: adamw
optim_conf:
    lr: 0.00001
    weight_decay: 0.000001
scheduler: warmuplr
scheduler_conf:
    warmup_steps: 6000
```

Several configurations were added to the original ESPnet. Each configuration is as follows:

- tau_ini / tau_end: The initial and final temperatures for Gumbel-Softmax.
- tau_cooldown_steps: The iteration number for the temperature of Gumbel-Softmax. Target sparsity gradually increases to sparsity_end.
- sparsity_init / sparsity_end : The initial target sparsity and after the warmup.
- sparsity_warmup_steps: Warmup steps for the sparsity. The target sparsity will be gradually increased to reach sparsity_end.

The class we set for the encoder and decoder is the extended class of E-Branchformer and TransformerDecoder to incorporate pruning in this study. Other configurations are same as the OWSM-v3.1, and you can refer to all settings in huggingface hub [2]

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

## A  PROBLEM FORMULATION OF DYNAMIC PRUNING WITH STRAIGHT-THROUGH GUMBEL-SOFTMAX ESTIMATOR

Our task formulation follows (Louizos et al., 2018), but differs in the formulation of the term used to compute model complexity, where we employ the SGSE function. Specifically, given an input speech signal $x$, model parameters $\theta$, model function $f(\theta)$, and pruning mask $z$, the output token $\tilde{y}$ is computed using the pruned parameters $\tilde{\theta}$ as follows:

$$\tilde{y} = f(x|\tilde{\theta}), \quad \tilde{\theta} = \theta \odot z, \quad z \in \{0,1\}^{|\theta|},$$

where $|\theta|$ denotes the number of model parameters. Given a dataset, we define the loss function $\mathcal{L}$ and the $l_0$ norm as $\|\tilde{\theta}\|_0 = \mathbb{E}[z]$. The training loss is then formulated as:

$$\text{loss} = \frac{1}{N}\left(\sum_{}^{N}\mathcal{L}(f(x|\tilde{\theta}),\ y)\right) + \lambda\|\tilde{\theta}\|_0,$$

where $N$ is the batch size and $\lambda$ is a hyperparameter that encourages model sparsity. Here, $\|\tilde{\theta}\|_0$ serves as a sparsity-inducing regularization term. However, learning $z$ as a binary variable makes it difficult to compute gradients for the skipped modules, which prevents parameter updates. To address this, (Louizos et al., 2018) applies a reparameterization trick to represent $z$ in a differentiable form $\tilde{z}$, allowing for simultaneous optimization of both $\theta$ and $\tilde{z}$ by defining $\|\tilde{\theta}\|_0 = \mathbb{E}[\tilde{z}]$.

However, this method results in $\tilde{z}$ being a continuous value, which leads to discrepancies between training and inference computations. To mitigate this issue, we employ SGSE, which ensures that $\tilde{z}$ remains binary even during training. Specifically, given a trainable global gate predictor $G(\cdot)$ and context information $C$, $\tilde{z}$ can be expressed using the SGSE function as:

$$\tilde{z} = \texttt{SGSE}(G(x, C))$$

## B  DATA SIZE

The language-specific data volumes used for training OWSM-v3.1 are shown in Table 2, including additional languages from the Appendix experiments. Although the Italian dataset is smaller than German and French, pruning trends remain similar. For performance details, see Appendix H. We created a new dataset by integrating Voxforge (Voxforge.org) with Europarl-ST, covering the same

Table 2: Data size used in OWSM-v3.1 pretraining for each language.

| Language | amount (h) |
|----------|-----------|
| French | 2489 |
| German | 3704 |
| Italian | 707 |
| Hungarian | 97 |
| Bulgarian | 18 |

Table 3: Additional dataset with Voxforge. We show WER for German (deu), French (fra), and Italian (ita). Decoder was pruned from OWSM-v3.1 model.

| sparsity | deu | fra | ita |
|----------|------|------|------|
| 0% | 14.2 | 11.1 | 12.3 |
| 10% | 14.8 | 12.8 | 13.3 |
| 30% | 15.1 | 12.9 | 13.6 |
| 50% | 19.3 | 16.0 | 18.2 |
| 70% | 17.5 | 15.1 | 15.8 |
| 90% | 25.5 | 23.8 | 21.6 |

Table 4: Additional results for Hungarian (hun) and Bulgarian (bul). We show WER for these two languages when encoders and decoders were pruned separately. The WER for full fine-tuned model is 33.2 for Hungarian and 23.5 for Bulgarian.

|          | Encoder | | Decoder | |
|----------|---------|------|---------|------|
| sparsity | hun | bul | hun | bul |
| 10% | 38.7 | 38.2 | 35.1 | 29.5 |
| 30% | 42.5 | 37.7 | 34.1 | 24.2 |
| 50% | 42.1 | 38.3 | 40.1 | 29.1 |
| 70% | 54.4 | 44.2 | 36.3 | 26.5 |
| 90% | 89.2 | 78.8 | 42.1 | 36.2 |

Table 5: Reproduced I3D results using the LibriSpeech dataset. We show the Word Error Rate (WER) on the test-clean set for varying numbers of encoder layers.

| # Layers | WER |
|----------|-----|
| 36 | 8.7 |
| 34 | 8.7 |
| 28 | 8.8 |

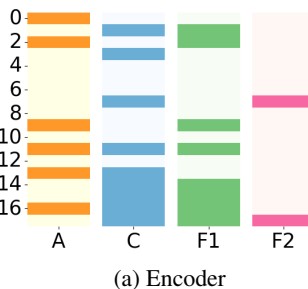

(a) Encoder

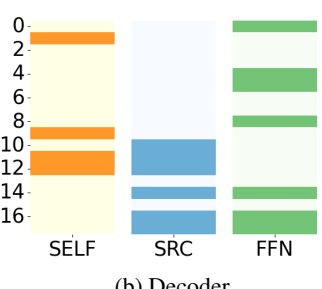

(b) Decoder

Figure 11: The sparsity plot for Hungarian ASR shows the results when the encoder and decoder were pruned separately, with the target sparsity ratio set to 70%. The pruning patterns exhibit no notable differences compared to those observed in high-resource languages.

languages. Training and evaluation follow Section 4, and ASR results appear in Table 3. Performance degradation across sparsity levels is comparable between Europarl-ST and the joint dataset.

To assess the impact on underrepresented languages, we selected Hungarian and Bulgarian from Fleurs (Conneau et al., 2022). Table 4 shows WER results. Decoder pruning lowered accuracy, as with high-resource languages like French and German, but encoder pruning had a greater impact. Even at lower sparsity, encoder pruning caused notable performance drops, highlighting its crucial role in preserving accuracy for low-resource languages. With less training data, effective input feature capture is essential, while the decoder is less sensitive to sparsity due to its reliance on encoder representations.

We developed a baseline on Peng et al. (2023b) using a Transformer-encoder model trained on LibriSpeech (Panayotov et al., 2015) and fine-tuned it on the LibriSpeech-100h subset. The reproduced WER results on the test-clean set are shown in Table 5. Our parameter settings yielded sparsity levels of approximately 95% and 78%, indicating that performance remains unaffected up to around 20–25% sparsity, consistent with our experimental results.

## C  LAYER-LEVEL PRUNING

To compare performance with other pruning methods, we conducted layer-level pruning experiments. An example pruning pattern for layer skipping is shown in Figure 12. Table 6 presents WER results for German ASR with decoder-only pruning. Up to 70% sparsity, skipping at the layer and module levels shows no significant difference. At lower sparsity, the layer-level approach performs better, while at higher sparsity, the module-level approach excels. This suggests that at higher sparsity, optimizing the roles and order of individual modules is more effective than skipping entire

Table 6: WER on module-level pruning and layer-level pruning. Results from German ASR, with only decoder pruned (left) and only encoder pruned (right).

| sparsity | Encoder Pruned | | Decoder Pruned | |
|---|---|---|---|---|
| | Module | Layer | Module | Layer |
| 10% | 14.5 | 31.3 | 15.8 | 15.4 |
| 30% | 15.0 | 32.0 | 16.4 | 15.2 |
| 50% | 16.8 | 34.2 | 16.2 | 16.5 |
| 70% | 25.1 | 40.2 | 18.3 | 18.7 |
| 90% | 47.3 | 108.6 | 26.4 | 99.9 |

Table 7: Comparison of GFLOPs between module skip and layer skip at different target sparsity levels when the encoder is pruned. We used the `fvcore` library to calculate the FLOPs. FLOPs were computed over multiple utterances, and the average value was taken. For reference, the FLOPs for the model without pruning were 3781.

| sparsity | Module Skip (GFLOPs) | Layer Skip (GFLOPs) |
|---|---|---|
| 10% | 3697 | 3697 |
| 30% | 3690 | 3692 |
| 50% | 3669 | 3666 |
| 70% | 3633 | 3640 |
| 90% | 3613 | 3620 |

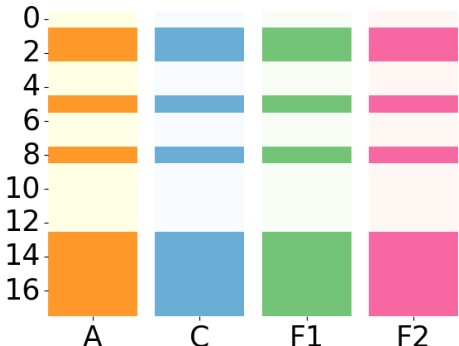

Figure 12: The sparsity plot when layers were skipped from encoder with the target sparsity ratio set to 70%. This model was trained on German-to-Italian ST.

layers. For the encoder, accuracy degradation is significant, suggesting that layer-level pruning is unsuitable for speech foundation models. Unlike module-level pruning, layer-level pruning forcibly removes even cgMLP, which plays a critical role in encoding.

We also computed FLOPs when the encoder is pruned, as shown in Table 7. Since different modules are used in module-level vs. layer-level pruning, we analyzed the impact. For encoder pruning, module-level pruning results in slightly lower FLOPs, reinforcing its advantage in efficiency.

## D  UNSTRUCTURED PRUNING

We investigated the impact of pruning strategies: unstructured vs. structured pruning. Table 8 compares magnitude-based unstructured pruning and PARP. Fine-tuning was performed on a dataset including ASR and ST, and WER was evaluated on German ASR. Even with unstructured pruning, decoder pruning preserved performance. However, compared to module-level pruning at the same sparsity, module-level pruning—especially on the encoder—maintained accuracy more effectively.

Table 8: WER results for German ASR with different unstructured pruning techniques.

| Model | WER (%) |
|---|---|
| Original model (without fine-tuning) | 24.6 |
| + Unstructured pruning (sparsity = 0.1) | 24.5 |
| + Unstructured pruning (sparsity = 0.3) | 24.5 |
| + PARP (sparsity = 0.1, applied to encoder) | 31.2 |
| + PARP (sparsity = 0.3, applied to encoder) | 31.8 |
| + PARP (sparsity = 0.1, applied to decoder) | 19.8 |
| + PARP (sparsity = 0.3, applied to decoder) | 16.4 |

## E  SPARSITY LOSS

In Peng et al. (2023b), pruning is applied to the Transformer encoder by targeting the self-attention and feed-forward network modules. The sparsity loss function, $\mathcal{L}_{\text{sparsity}}$, is defined as:

$$\mathcal{L}_{\text{sparsity}} = \lambda \left( \frac{1}{2N} \sum_{l=1}^{N} (g_{\text{self-ATT}}^{(l)} + g_{\text{FFN}}^{(l)}) \right),$$

where $g_{\text{self-ATT}}^{(l)}, g_{\text{FFN}}^{(l)}$ are the gate probabilities of each module in the $l$-th layer, and the $N$ is the number of layers. $\lambda$ is a constant loss weight for the sparsity and the value is determined heuristically. Peng et al. (2023b) aims to achieve model sparsity by controlling the magnitude of the loss through $\lambda$. In Peng et al. (2023b), different values of $\lambda$ will lead to different inference costs, as the final sparsity ratio of the model is controlled by sparsity loss.

We initially tried the fixed value for $\lambda$. However, the fixed $\lambda$ did not allow the model to achieve the desired sparsity, particularly when attempting to prune over 70% of the modules. To address this, Wang et al. (2020) introduces a Lagrange multiplier $\lambda_1$ and $\lambda_2$, and the sparsity loss is defined as:

$$\mathcal{L}_{penalty} = \lambda_1 (g - s_{\text{target}}) + \lambda_2 (g - s_{\text{target}})^2$$

where $\lambda_1$ and $\lambda_2$ are updated based on the model's sparsity.

Here, $\mathcal{L}_{penalty}$ can take a negative value when $-\frac{\lambda_2}{\lambda_1} \le g - s_{\text{target}} \le 0$ (Details are in F). If $\mathcal{L}_{penalty}$ becomes negative, it complicates solving the minimization problem when combined with the ASR and ST losses. To address this issue, we employed a function to calculate the absolute value of $g - s_{\text{target}}$, so that the $\mathcal{L}_{penalty}$ remains non-negative. For the sake of simplicity, we set the $\lambda_1 = \lambda_2 = 1$ and employed a constant $\alpha$ on top of it. Starting from 1, we gradually increase the $\alpha$ unless it reaches the desired sparsity. The visualization of each cost function is in the Appendix. Thus, the sparsity loss we used in this experiment becomes Eq. 1. When calculating the $\mathcal{L}_{\text{sparsity}}$, $\alpha$ was gradually increased following Wang et al. (2020).

## F  VISUALIZATION ON SPARSITY LOSS

The purpose of Lagrange multipliers in Wang et al. (2020) is to make the $\mathcal{L}_{\text{penalty}}$ more aggressive penalty term. So we hypothesized that simply introducing a regularization term, which has similar role, we can make the model prune the desired number of modules. Figure 13 illustrates the graphs of various penalty terms. The x-axis represents $g_{l_0} - s_{\text{target}}$ and the y-axis represents $\mathcal{L}_{\text{sparsity}}$.

The $\mathcal{L}_{\text{penalty}}$ in Appendix E can be rewritten as a quadratic function: $\mathcal{L}_{\text{penalty}} = (x + \frac{\lambda_1}{2\lambda_2})^2 - \frac{\lambda_2^2}{4\lambda_1^2}$ This is represented by the green curve in Figure 13. They move the vertex of this green curve when updating the two Lagrange multipliers. In our implementation, we set $\lambda_1 = \lambda_2 = 1$ and introduced an overall coefficient $\alpha$. Thus, $\mathcal{L}_{\text{penalty}}$ becomes: $\mathcal{L}_{\text{penalty}} = \alpha(x + \frac{1}{2})^2 - \frac{\alpha}{4}$ In our approach, we increase the value of this coefficient when the difference between the actual sparsity and the target sparsity exceeds a certain threshold (0.05). Increasing this coefficient $\alpha$ lowers the vertex of the green line in Figure 13. For example, setting $\alpha = 5$ results in the red curve, which has a steeper slope for $x > 0$.

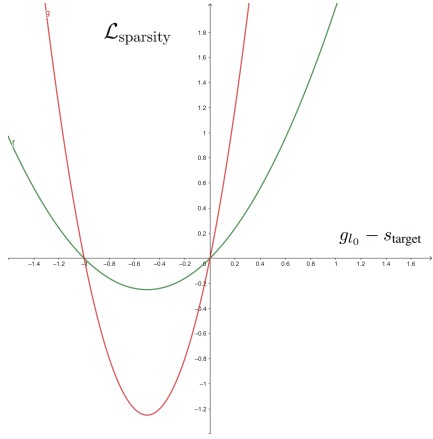

Figure 13: Visualization on different $\mathcal{L}_{\text{sparsity}}$. Wang et al. (2020) moves the green line to maximize the loss. Our method simply shifts the vertex downward. For example, by lowering the vertex of the green line, we transform it into a penalty function resembling the red line.

---

**Algorithm 1** Gate Predictor

---

$x_{\text{pooled}} \leftarrow \text{Average}(x_{\text{GP}})$      $\triangleright$ Average over time dimension, $x_{\text{pooled}} \in \mathbb{R}^D$
$x \leftarrow \text{Concat}(C, x_{\text{pooled}})$      $\triangleright$ Concatenate conditional info, $x \in \mathbb{R}^{D+D_{\text{conf}}}$
$\text{logit} \leftarrow \text{Reshape}(G(x))$      $\triangleright$ Reshape $G(x)$ to logit $\in \mathbb{R}^{L \times 2}$
$g \leftarrow \text{SGSE}(\text{logit}, \text{axis} = 1)$      $\triangleright$ Compute SGSE function
$g \leftarrow g[:, 1]$      $\triangleright$ Select second column from Gumbel-Softmax output

---

## G  CONTEXT-AWARE GATE PROBABILITY

The gate probability $g$ is computed using a separate predictor for each module. Let $G()$ denote the gate predictor, with speech feature input $x_{\text{GP}} \in \mathbb{R}^{T \times D}$, condition $C \in \mathbb{R}^{D^C}$, and total layers $L$. Here, $T$ is the number of frames, $D$ the speech feature dimension, and $D^C$ the condition dimension. The gate probability $g$ for a module is computed as in Algorithm 1, with $g_{l_0} = \mathbb{E}[g]$.

To prevent early learning instability, we avoid random weight initialization for the Gate Predictor. Random initialization may remove 50% of modules early, hindering gradual reduction. Instead, we adjust the final layer bias so that $g$ initially outputs values near 1, ensuring full activation. This setup allows a gradual reduction in active modules during training. Following Peng et al. (2023b), we use a two-layer MLP with an intermediate size of 32.

## H  HEATMAPS AND TABLES

Table 9: Comparison of WER (%) for French, German, and Italian using encoder-sparsified model, decoder-sparsified model, and jointly sparsified model.

| | Baseline | Sparse Encoder | | | | | Sparse Decoder | | | | | Jointly Sparsified Encoder-Decoder | | | | |
|---|---|---|---|---|---|---|---|---|---|---|---|---|---|---|---|---|
| | | 0.1 | 0.3 | 0.5 | 0.7 | 0.9 | 0.1 | 0.3 | 0.5 | 0.7 | 0.9 | 0.1 | 0.3 | 0.5 | 0.7 | 0.9 |
| French | 10.3 | 11.4 | 11.7 | 12.8 | 18.3 | 84.8 | 11.3 | 13.4 | 16.4 | 13.3 | 26.5 | 10.9 | 11.7 | 14.1 | 27.1 | 84.8 |
| German | 13.5 | 14.4 | 14.8 | 16.3 | 21.0 | 80.8 | 15.3 | 15.9 | 20.4 | 17.2 | 24.5 | 14.3 | 15.4 | 17.8 | 28.8 | 80.8 |
| Italian | 12.8 | 14.5 | 14.4 | 16.5 | 22.5 | 86.0 | 13.8 | 13.9 | 20.1 | 15.4 | 26.5 | 13.5 | 14.7 | 17.5 | 32.5 | 86.0 |

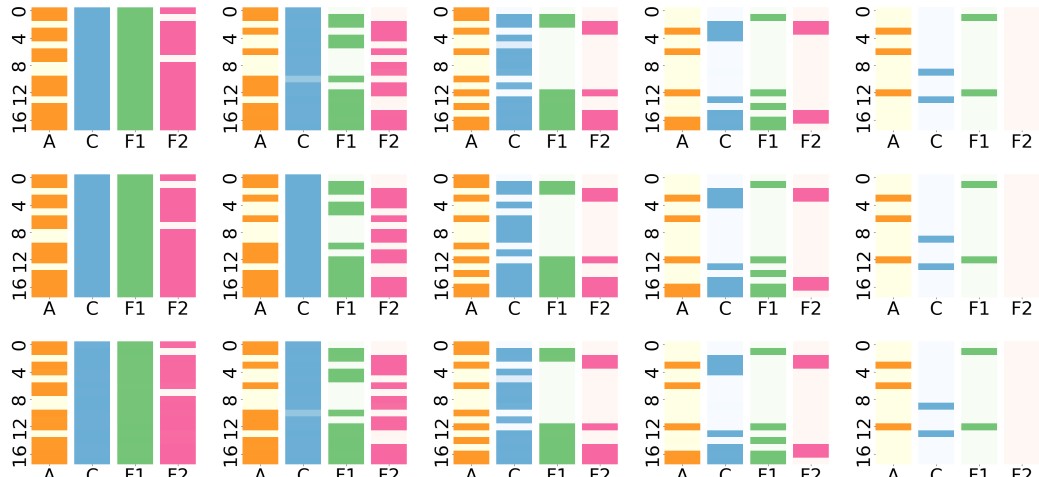

Figure 14: Visualization of $\mathbb{E}[g_{\text{enc}}]$ when encoder was pruned separately. The columns represents the $s_{\text{target}}$, with $s_{\text{target}}$ being 0.1, 0.3, 0.5, 0.7, and 0.9 from left to right. The first row represents the French, the second row represents German, and the third row represents Italian. The other settings are consistent with those in Figure 3.

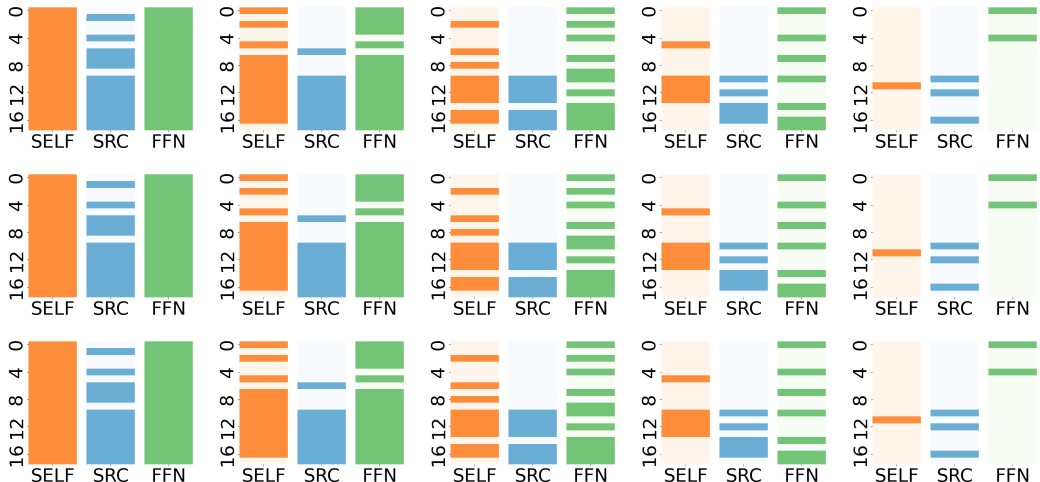

Figure 15: Visualization of $\mathbb{E}[g_{\text{dec}}]$ when decoder was pruned separately. The columns represents the $s_{\text{target}}$, with $s_{\text{target}}$ being 0.1, 0.3, 0.5, 0.7, and 0.9 from left to right. The first row represents the French, the second row represents German, and the third row represents Italian. The other settings are consistent with those in Figure 3.

Table 10: BLEU score on each speech translation direction for Sparse Encoder, Sparse Decoder, and Jointly Sparsified Encoder-Decoder

| src | trg | baseline | Sparse Encoder | | | | | Sparse Decoder | | | | | Jointly Sparsified Encoder-Decoder | | | | |
|-----|-----|----------|------|------|------|------|------|------|------|------|------|------|------|------|------|------|------|
| | | | 0.1 | 0.3 | 0.5 | 0.7 | 0.9 | 0.1 | 0.3 | 0.5 | 0.7 | 0.9 | 0.1 | 0.3 | 0.5 | 0.7 | 0.9 |
| fra | de | 12.2 | 12.3 | 11.9 | 12.3 | 9.2 | 2.5 | 12.9 | 13.0 | 12.2 | 10.3 | 5.3 | 12.2 | 11.6 | 9.4 | 5.2 | 0.4 |
| | it | 13.5 | 12.8 | 12.7 | 12.3 | 10.2 | 1.8 | 12.5 | 11.8 | 12.0 | 9.3 | 3.1 | 12.8 | 12.0 | 8.8 | 4.2 | 0.0 |
| deu | fr | 9.4 | 9.1 | 8.7 | 8.5 | 6.6 | 2.2 | 9.3 | 9.2 | 9.0 | 6.4 | 3.1 | 8.9 | 8.5 | 6.3 | 3.2 | 3.1 |
| | it | 7.5 | 7.0 | 7.0 | 6.6 | 5.1 | 1.3 | 8.4 | 8.4 | 7.9 | 6.4 | 2.8 | 7.1 | 6.5 | 4.7 | 2.3 | 0.0 |
| ita | de | 11.8 | 11.1 | 10.8 | 10.1 | 8.2 | 2.8 | 12.1 | 12.0 | 11.8 | 9.4 | 5.5 | 11.2 | 10.3 | 8.5 | 4.6 | 0.7 |
| | fr | 14.0 | 12.8 | 12.6 | 11.8 | 9.9 | 3.1 | 13.0 | 12.3 | 11.8 | 8.3 | 3.3 | 12.8 | 12.0 | 8.8 | 4.2 | 0.9 |

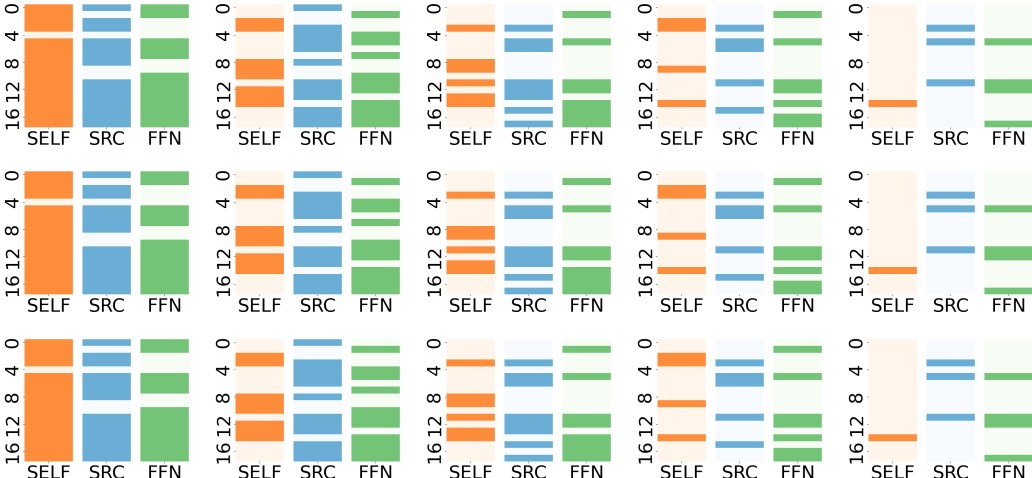

Figure 16: Visualization of $\mathbb{E}[g_{\text{enc}}]$ when encoder and decoder was pruned jointly. The columns represents the $s_{\text{target}}$, with $s_{\text{target}}$ being 0.1, 0.3, 0.5, 0.7, and 0.9 from left to right. The first row represents the French, the second row represents German, and the third row represents Italian. The other settings are consistent with those in Figure 3.

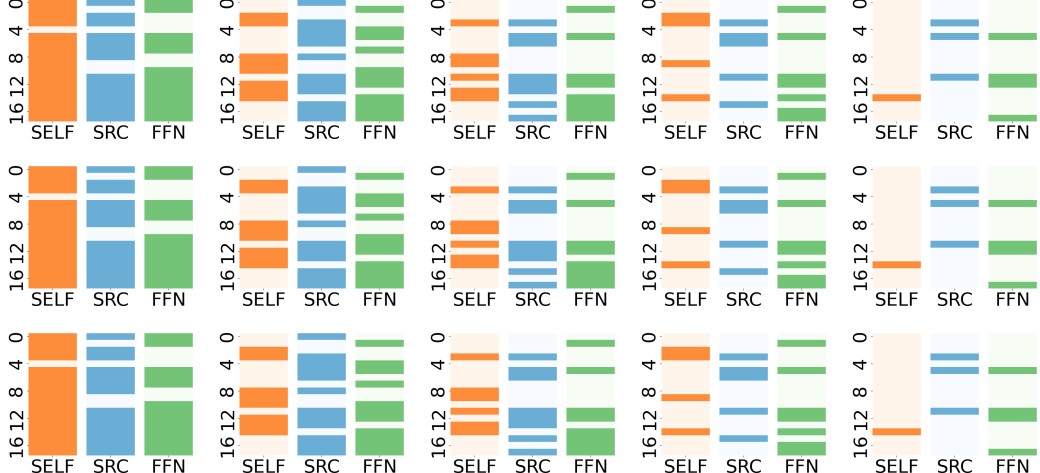

Figure 17: Visualization of $\mathbb{E}[g_{\text{dec}}]$ when encoder and decoder was pruned jointly. The columns represents the $s_{\text{target}}$, with $s_{\text{target}}$ being 0.1, 0.3, 0.5, 0.7, and 0.9 from left to right. The first row represents the French, the second row represents German, and the third row represents Italian. The other settings are consistent with those in Figure 3.

Table 11: BLEU score on each speech translation direction and WER for ASR tasks for Sparse Encoder, Sparse Decoder, and Jointly Sparsified Encoder-Decoder. The leftmost column represents the source speech's language and the corresponding target text's language.

| src-trg | metric | baseline | Sparse Encoder | | | | | Sparse Decoder | | | | | Jointly Sparsified Encoder-Decoder | | | | |
|---|---|---|---|---|---|---|---|---|---|---|---|---|---|---|---|---|---|
| | | | 0.1 | 0.3 | 0.5 | 0.7 | 0.9 | 0.1 | 0.3 | 0.5 | 0.7 | 0.9 | 0.1 | 0.3 | 0.5 | 0.7 | 0.9 |
| fr-fr | WER | 11.0 | 11.6 | 11.9 | 13.1 | 22.3 | 46.8 | 12.6 | 13.4 | 13.2 | 15.0 | 25.9 | 11.2 | 12.4 | 14.5 | 24.2 | 107.0 |
| fr-de | BLEU | 11.2 | 8.4 | 8.6 | 7.9 | 3.9 | 2.0 | 11.3 | 11.6 | 10.2 | 9.0 | 5.8 | 10.7 | 9.9 | 8.3 | 5.5 | 0.6 |
| fr-it | BLEU | 13.0 | 12.2 | 12.3 | 11.6 | 7.6 | 2.5 | 12.3 | 11.9 | 11.2 | 9.6 | 4.7 | 10.2 | 9.5 | 7.8 | 2.9 | 0.0 |
| de-de | WER | 13.6 | 14.5 | 15.0 | 16.8 | 25.1 | 47.3 | 15.8 | 16.4 | 16.2 | 18.3 | 26.4 | 14.6 | 15.0 | 18.2 | 26.5 | 105.0 |
| de-fr | BLEU | 8.4 | 6.1 | 5.8 | 5.0 | 3.1 | 1.5 | 8.2 | 8.3 | 7.7 | 6.3 | 4.0 | 8.0 | 7.3 | 6.0 | 3.8 | 0.4 |
| de-it | BLEU | 6.4 | 6.1 | 6.0 | 5.7 | 3.4 | 1.2 | 6.9 | 7.0 | 6.2 | 4.9 | 2.6 | 4.9 | 4.7 | 3.7 | 1.7 | 0.0 |
| it-it | WER | 13.2 | 14.4 | 15.2 | 17.3 | 27.6 | 86.0 | 14.5 | 15.2 | 15.0 | 16.7 | 28.9 | 13.9 | 15.1 | 17.8 | 28.7 | 162.6 |
| it-de | BLEU | 9.8 | 6.5 | 6.5 | 6.2 | 3.9 | 2.2 | 10.6 | 10.4 | 9.8 | 8.4 | 5.8 | 7.1 | 8.7 | 7.4 | 3.2 | 0.1 |
| it-fr | BLEU | 13.5 | 13.0 | 12.6 | 11.9 | 8.1 | 2.9 | 12.6 | 12.6 | 11.9 | 9.5 | 5.6 | 12.6 | 11.5 | 9.9 | 5.6 | 0.6 |

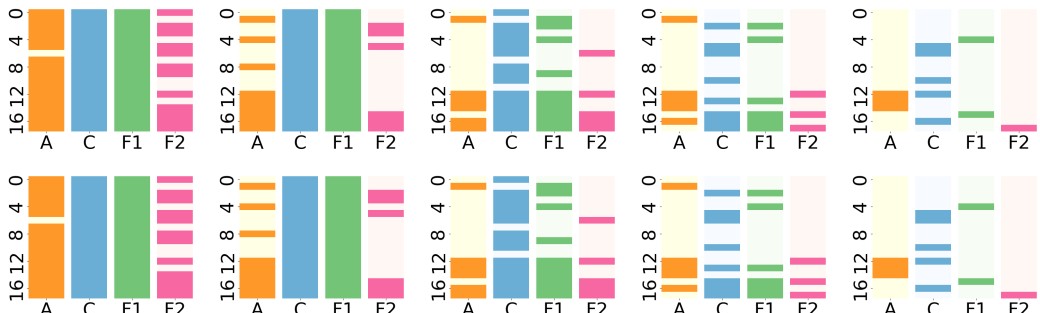

Figure 18: Visualization of $\mathbb{E}[g_{\text{enc}}]$ when encoder was pruned separately. The columns represents the $s_{\text{target}}$, with $s_{\text{target}}$ being 0.1, 0.3, 0.5, 0.7, and 0.9 from left to right. The first row represents the French-to-German, the second row represents French-to-Italian translation. The other settings are consistent with those in Figure 3.

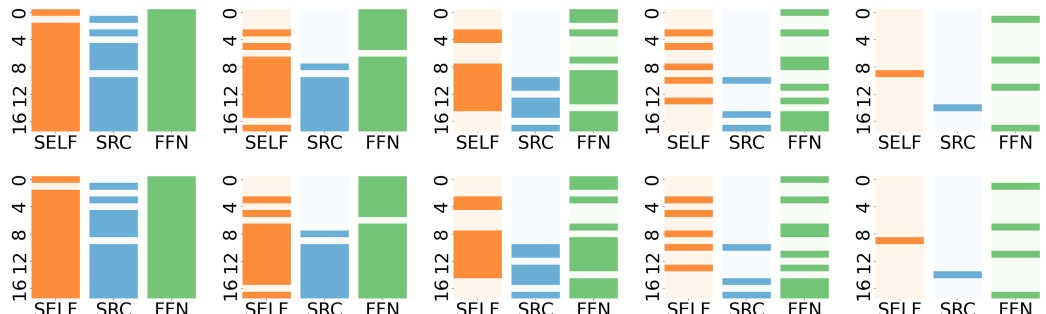

Figure 19: Visualization of $\mathbb{E}[g_{\text{dec}}]$ when decoder was pruned separately. The columns represents the $s_{\text{target}}$, with $s_{\text{target}}$ being 0.1, 0.3, 0.5, 0.7, and 0.9 from left to right. The first row represents the French-to-German, the second row represents French-to-Italian translation. The other settings are consistent with those in Figure 3.

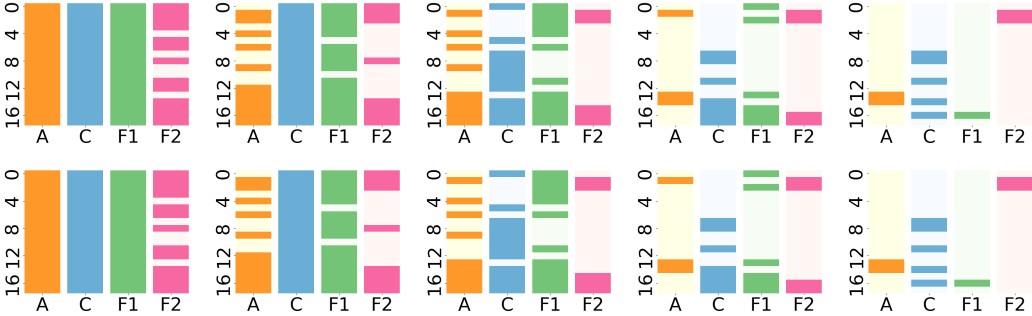

Figure 20: Visualization of $\mathbb{E}[g_{\text{enc}}]$ when encoder was jointly pruned with decoder. The columns represents the $s_{\text{target}}$, with $s_{\text{target}}$ being 0.1, 0.3, 0.5, 0.7, and 0.9 from left to right. The first row represents the French-to-German, the second row represents French-to-Italian translation. The other settings are consistent with those in Figure 3.

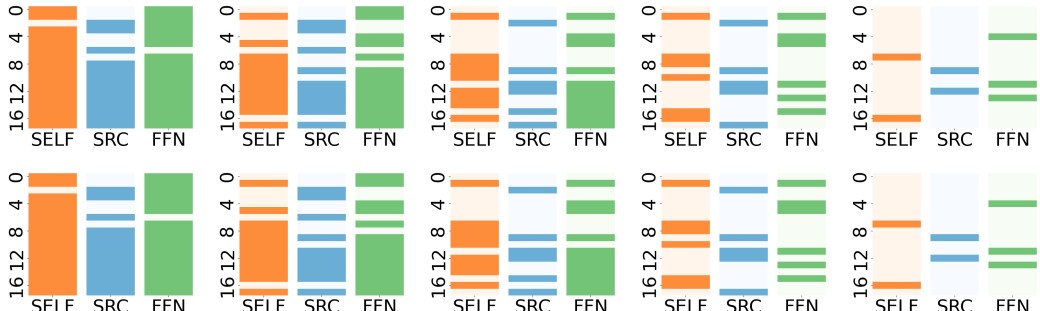

Figure 21: Visualization of $\mathbb{E}[g_{\text{dec}}]$ when decoder was jointly pruned with encoder. The columns represents the $s_{\text{target}}$, with $s_{\text{target}}$ being 0.1, 0.3, 0.5, 0.7, and 0.9 from left to right. The first row represents the French-to-German, the second row represents French-to-Italian translation. The other settings are consistent with those in Figure 3.

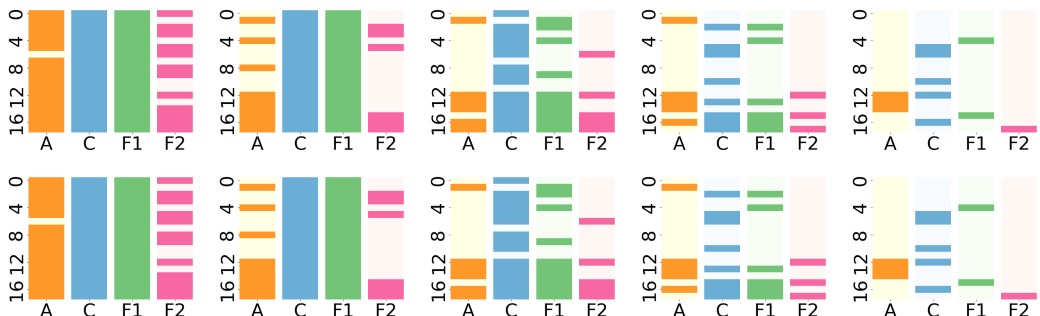

Figure 22: Visualization of $\mathbb{E}[g_{\text{enc}}]$ when encoder was pruned separately. The columns represents the $s_{\text{target}}$, with $s_{\text{target}}$ being 0.1, 0.3, 0.5, 0.7, and 0.9 from left to right. The first row represents the German-to-French, the second row represents German-to-Italian translation. The other settings are consistent with those in Figure 3.

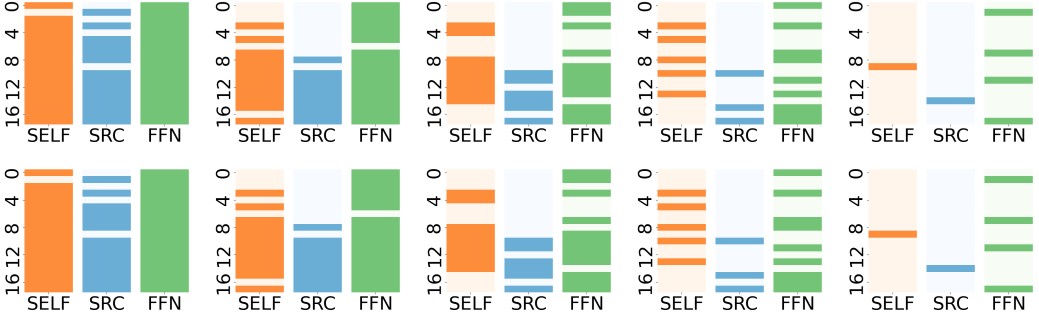

Figure 23: Visualization of $\mathbb{E}[g_{\text{dec}}]$ when decoder was pruned separately. The columns represents the $s_{\text{target}}$, with $s_{\text{target}}$ being 0.1, 0.3, 0.5, 0.7, and 0.9 from left to right. The first row represents the German-to-French, the second row represents German-to-Italian translation. The other settings are consistent with those in Figure 3.

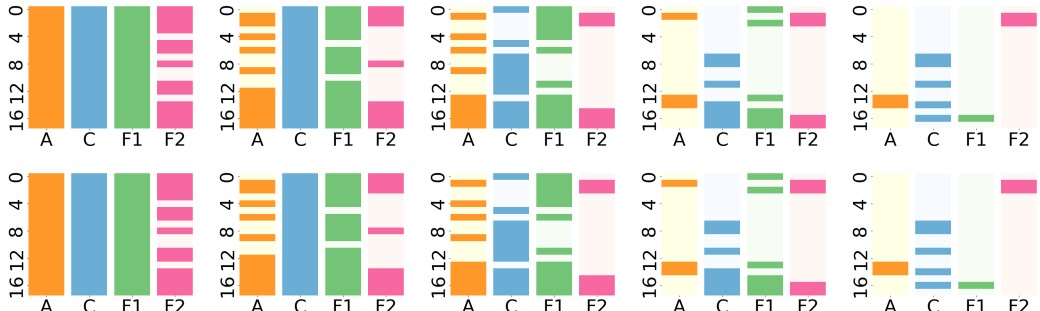

Figure 24: Visualization of $\mathbb{E}[g_{\text{enc}}]$ when encoder was jointly pruned with decoder. The columns represents the $s_{\text{target}}$, with $s_{\text{target}}$ being 0.1, 0.3, 0.5, 0.7, and 0.9 from left to right. The first row represents the German-to-French, the second row represents German-to-Italian translation. The other settings are consistent with those in Figure 3.

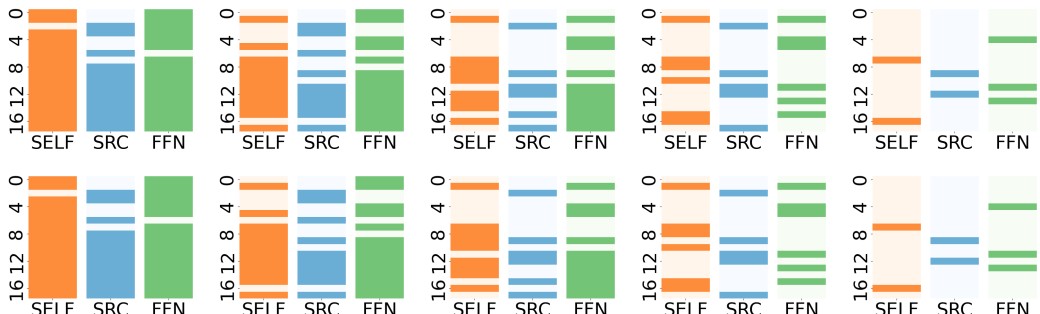

Figure 25: Visualization of $\mathbb{E}[g_{\text{dec}}]$ when decoder was jointly pruned with encoder. The columns represents the $s_{\text{target}}$, with $s_{\text{target}}$ being 0.1, 0.3, 0.5, 0.7, and 0.9 from left to right. The first row represents the German-to-French, the second row represents German-to-Italian translation. The other settings are consistent with those in Figure 3.

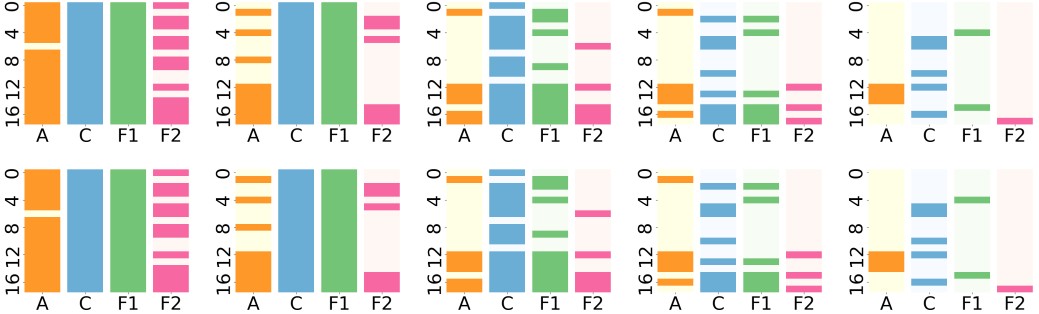

Figure 26: Visualization of $\mathbb{E}[g_{\text{enc}}]$ when encoder was pruned separately. The columns represents the $s_{\text{target}}$, with $s_{\text{target}}$ being 0.1, 0.3, 0.5, 0.7, and 0.9 from left to right. The first row represents the Italian-to-French, the second row represents Italian-to-German translation. The other settings are consistent with those in Figure 3.

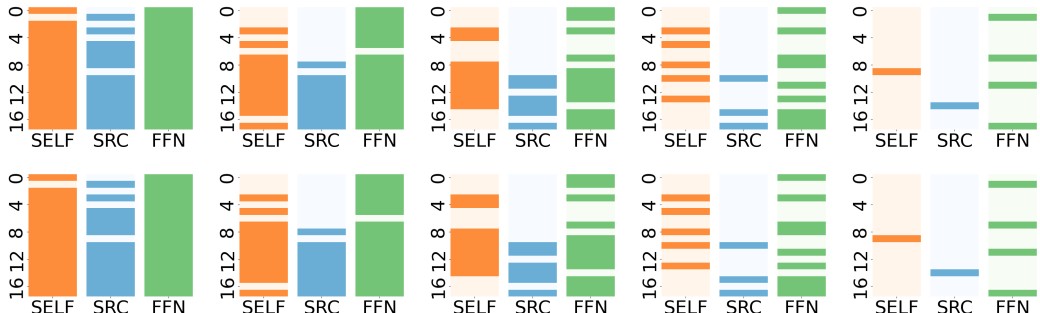

Figure 27: Visualization of $\mathbb{E}[g_{\text{dec}}]$ when decoder was pruned separately. The columns represents the $s_{\text{target}}$, with $s_{\text{target}}$ being 0.1, 0.3, 0.5, 0.7, and 0.9 from left to right. The first row represents the Italian-to-French, the second row represents Italian-to-German translation. The other settings are consistent with those in Figure 3.

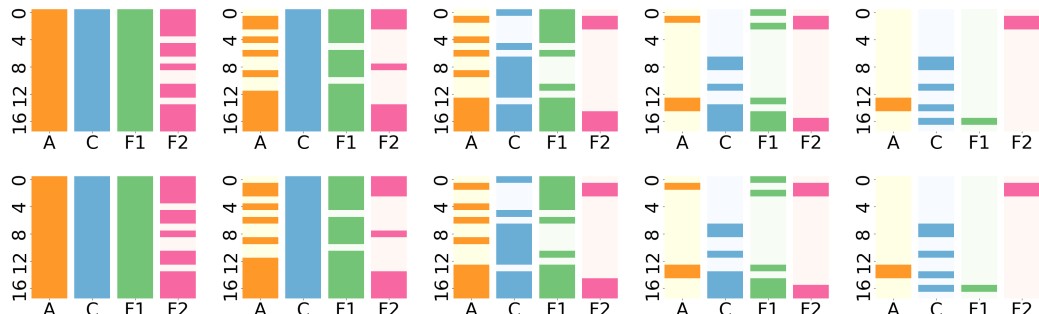

Figure 28: Visualization of $\mathbb{E}[g_{\text{enc}}]$ when encoder was jointly pruned with decoder. The columns represents the $s_{\text{target}}$, with $s_{\text{target}}$ being 0.1, 0.3, 0.5, 0.7, and 0.9 from left to right. The first row represents the Italian-to-French, the second row represents Italian-to-German translation. The other settings are consistent with those in Figure 3.

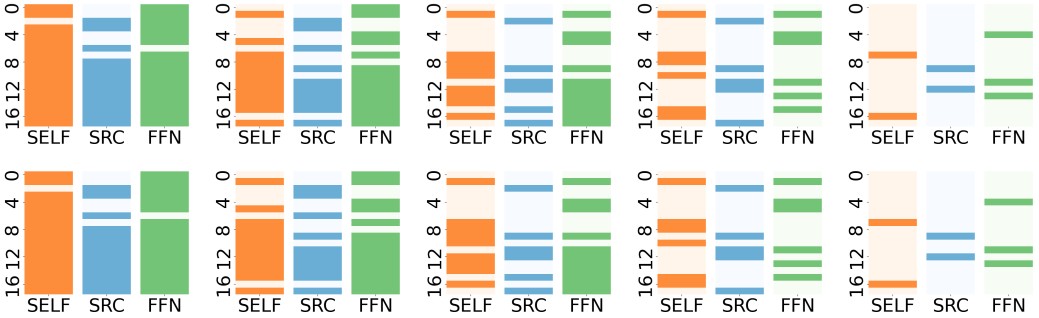

Figure 29: Visualization of $\mathbb{E}[g_{\text{dec}}]$ when decoder was jointly pruned with encoder. The columns represents the $s_{\text{target}}$, with $s_{\text{target}}$ being 0.1, 0.3, 0.5, 0.7, and 0.9 from left to right. The first row represents the Italian-to-French, the second row represents Italian-to-German translation. The other settings are consistent with those in Figure 3.

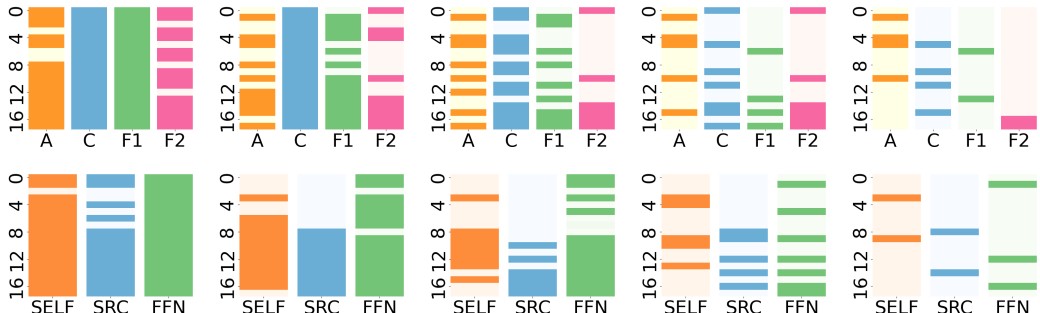

Figure 30: Visualization of $\mathbb{E}[g_{enc}]$ and $\mathbb{E}[g_{dec}]$ when encoder and decoder was pruned separately for French ASR. The columns represents the $s_{target}$, with $s_{target}$ being 0.1, 0.3, 0.5, 0.7, and 0.9 from left to right. The first row represents the encoder, the second row represents the decoder. The other settings are consistent with those in Figure 3.

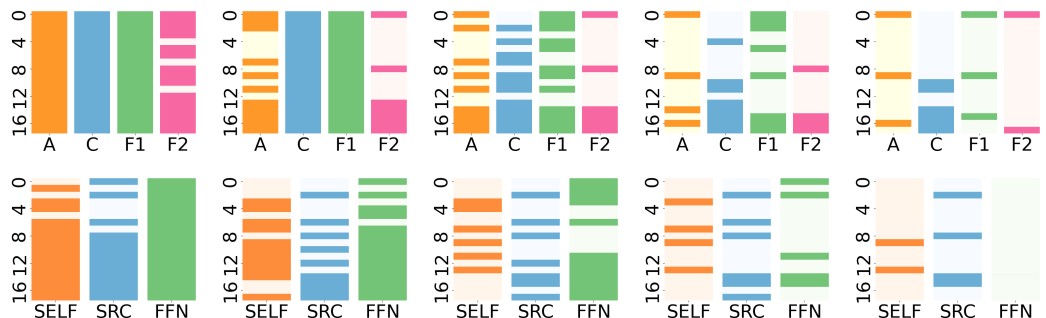

Figure 31: Visualization of $\mathbb{E}[g_{enc}]$ and $\mathbb{E}[g_{dec}]$ when encoder and decoder was pruned jointly for French ASR. The columns represents the $s_{target}$, with $s_{target}$ being 0.1, 0.3, 0.5, 0.7, and 0.9 from left to right. The first row represents the encoder, the second row represents the decoder. The other settings are consistent with those in Figure 3.

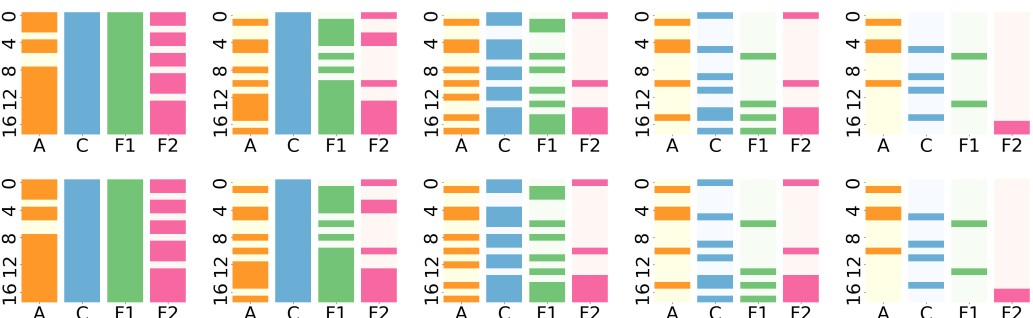

Figure 32: Visualization of $\mathbb{E}[g_{enc}]$ when encoder was pruned separately. The columns represents the $s_{target}$, with $s_{target}$ being 0.1, 0.3, 0.5, 0.7, and 0.9 from left to right. The first row represents the French-to-German, the second row represents French-to-Italian translation. The other settings are consistent with those in Figure 3.

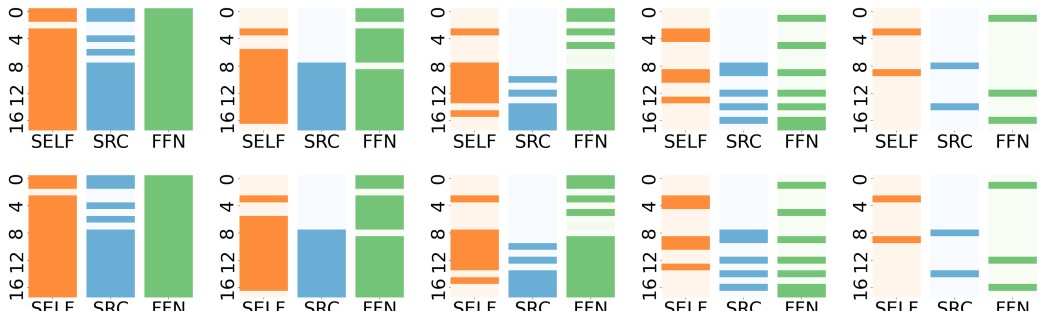

Figure 33: Visualization of $\mathbb{E}[g_{\text{dec}}]$ when decoder was pruned separately. The columns represents the $s_{\text{target}}$, with $s_{\text{target}}$ being 0.1, 0.3, 0.5, 0.7, and 0.9 from left to right. The first row represents the French-to-German, the second row represents French-to-Italian translation. The other settings are consistent with those in Figure 3.

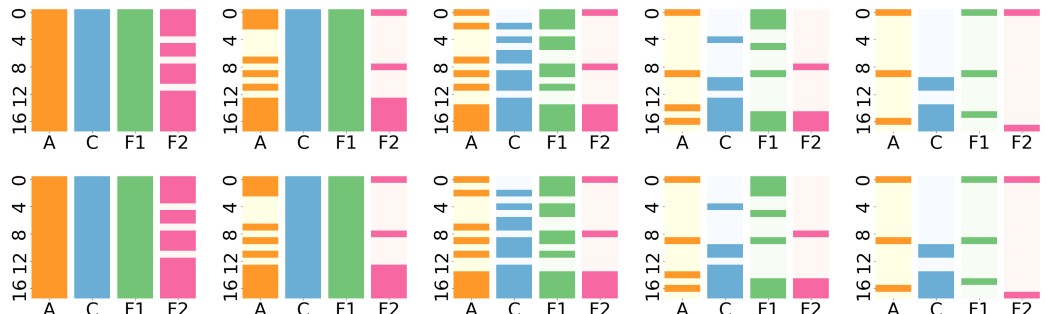

Figure 34: Visualization of $\mathbb{E}[g_{\text{enc}}]$ when encoder was jointly pruned with decoder. The columns represents the $s_{\text{target}}$, with $s_{\text{target}}$ being 0.1, 0.3, 0.5, 0.7, and 0.9 from left to right. The first row represents the French-to-German, the second row represents French-to-Italian translation. The other settings are consistent with those in Figure 3.

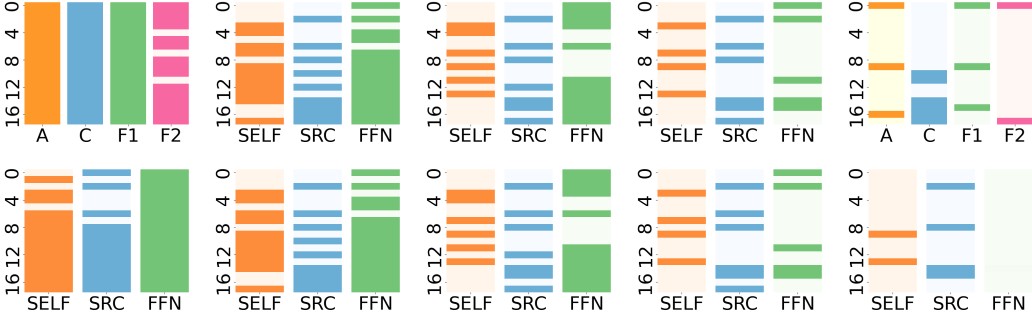

Figure 35: Visualization of $\mathbb{E}[g_{\text{dec}}]$ when decoder was jointly pruned with encoder. The columns represents the $s_{\text{target}}$, with $s_{\text{target}}$ being 0.1, 0.3, 0.5, 0.7, and 0.9 from left to right. The first row represents the French-to-German, the second row represents French-to-Italian translation. The other settings are consistent with those in Figure 3.

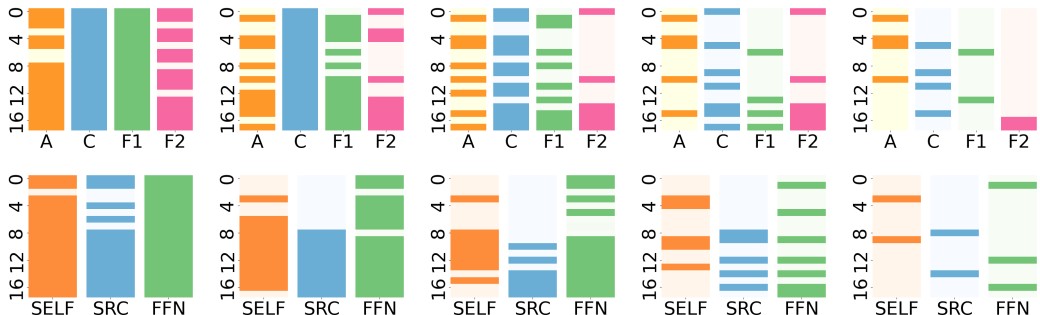

Figure 36: Visualization of $\mathbb{E}[g_{\text{enc}}]$ and $\mathbb{E}[g_{\text{dec}}]$ when encoder and decoder was pruned separately for German ASR. The columns represents the $s_{\text{target}}$, with $s_{\text{target}}$ being 0.1, 0.3, 0.5, 0.7, and 0.9 from left to right. The first row represents the encoder, the second row represents the decoder. The other settings are consistent with those in Figure 3.

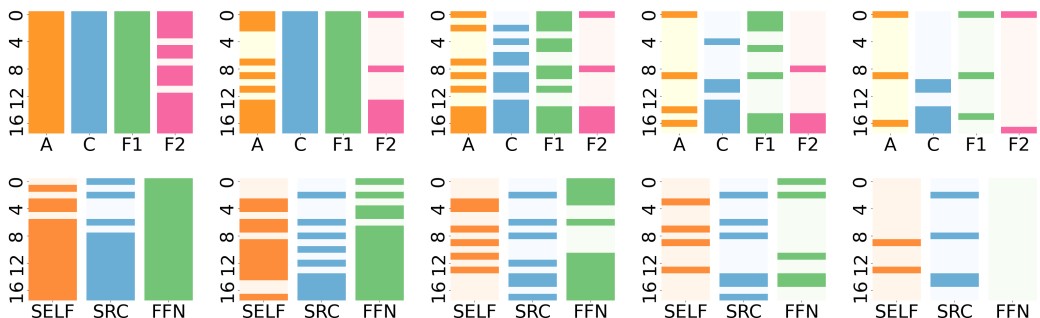

Figure 37: Visualization of $\mathbb{E}[g_{\text{enc}}]$ and $\mathbb{E}[g_{\text{dec}}]$ when encoder and decoder was pruned jointly for German ASR. The columns represents the $s_{\text{target}}$, with $s_{\text{target}}$ being 0.1, 0.3, 0.5, 0.7, and 0.9 from left to right. The first row represents the encoder, the second row represents the decoder. The other settings are consistent with those in Figure 3.

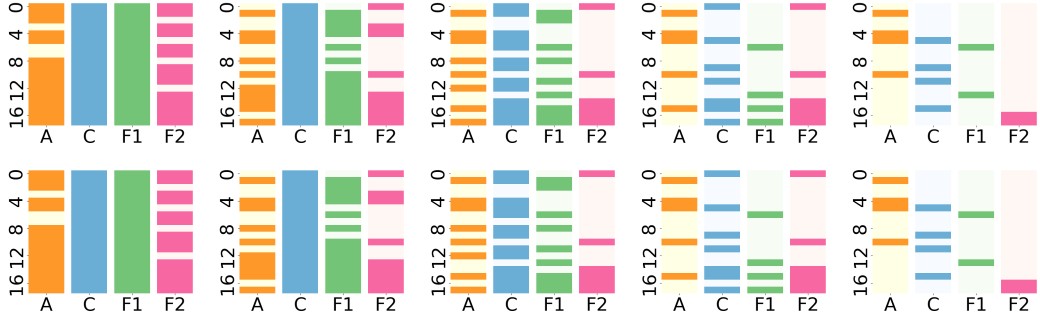

Figure 38: Visualization of $\mathbb{E}[g_{\text{enc}}]$ when encoder was pruned separately. The columns represents the $s_{\text{target}}$, with $s_{\text{target}}$ being 0.1, 0.3, 0.5, 0.7, and 0.9 from left to right. The first row represents the German-to-French, the second row represents German-to-Italian translation. The other settings are consistent with those in Figure 3.

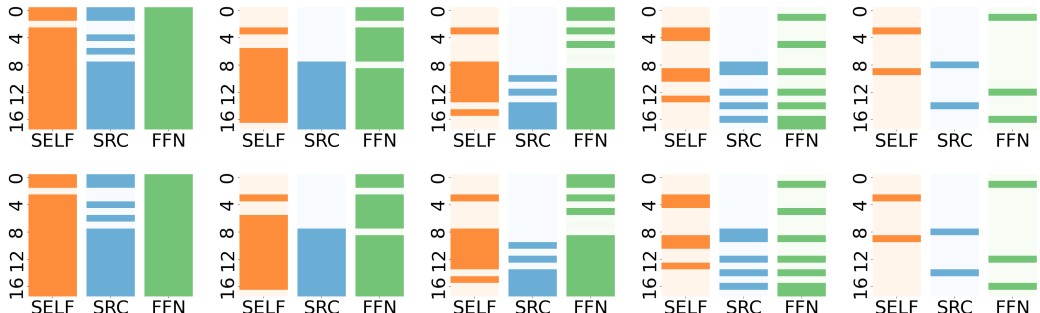

Figure 39: Visualization of $\mathbb{E}[g_{\text{dec}}]$ when decoder was pruned separately. The columns represents the $s_{\text{target}}$, with $s_{\text{target}}$ being 0.1, 0.3, 0.5, 0.7, and 0.9 from left to right. The first row represents the German-to-French, the second row represents German-to-Italian translation. The other settings are consistent with those in Figure 3.

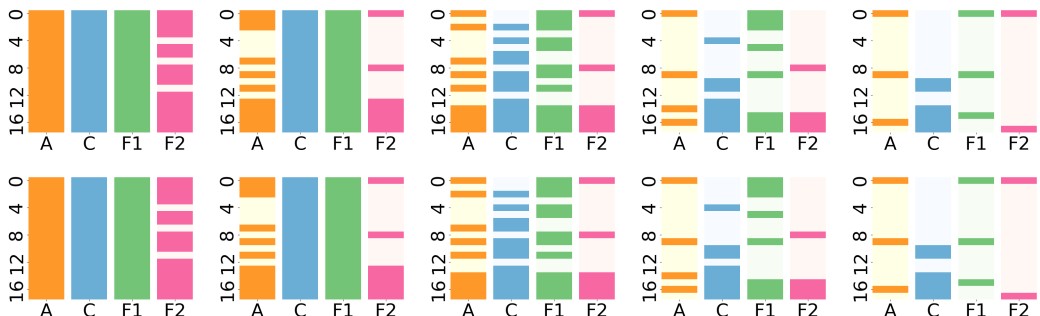

Figure 40: Visualization of $\mathbb{E}[g_{\text{enc}}]$ when encoder was jointly pruned with decoder. The columns represents the $s_{\text{target}}$, with $s_{\text{target}}$ being 0.1, 0.3, 0.5, 0.7, and 0.9 from left to right. The first row represents the German-to-French, the second row represents German-to-Italian translation. The other settings are consistent with those in Figure 3.

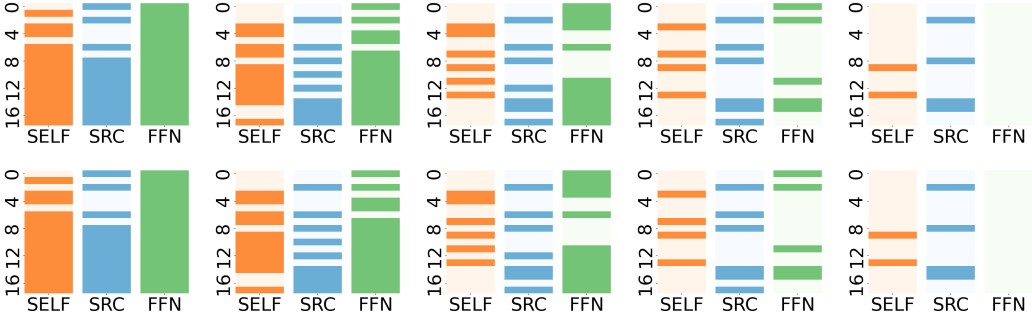

Figure 41: Visualization of $\mathbb{E}[g_{\text{dec}}]$ when decoder was jointly pruned with encoder. The columns represents the $s_{\text{target}}$, with $s_{\text{target}}$ being 0.1, 0.3, 0.5, 0.7, and 0.9 from left to right. The first row represents the German-to-French, the second row represents German-to-Italian translation. The other settings are consistent with those in Figure 3.

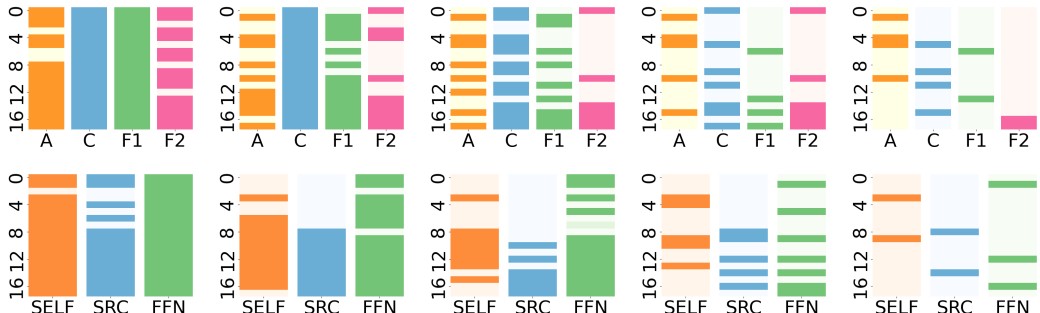

Figure 42: Visualization of $\mathbb{E}[g_{\text{enc}}]$ and $\mathbb{E}[g_{\text{dec}}]$ when encoder and decoder was pruned separately for Italian ASR. The columns represents the $s_{\text{target}}$, with $s_{\text{target}}$ being 0.1, 0.3, 0.5, 0.7, and 0.9 from left to right. The first row represents the encoder, the second row represents the decoder. The other settings are consistent with those in Figure 3.

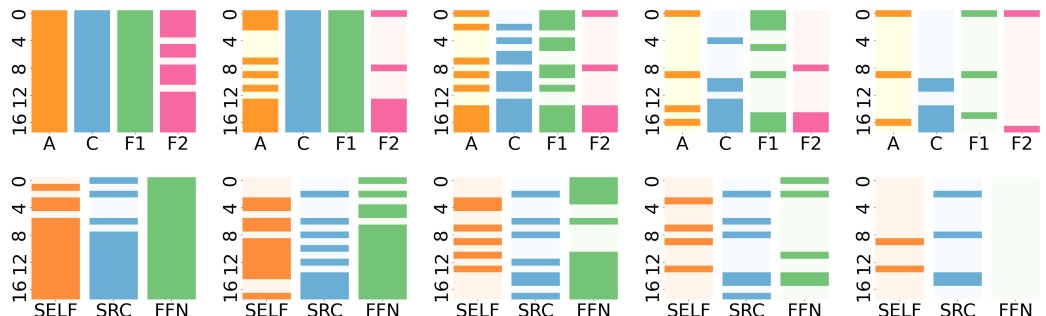

Figure 43: Visualization of $\mathbb{E}[g_{\text{enc}}]$ and $\mathbb{E}[g_{\text{dec}}]$ when encoder and decoder was pruned jointly for Italian ASR. The columns represents the $s_{\text{target}}$, with $s_{\text{target}}$ being 0.1, 0.3, 0.5, 0.7, and 0.9 from left to right. The first row represents the encoder, the second row represents the decoder. The other settings are consistent with those in Figure 3.

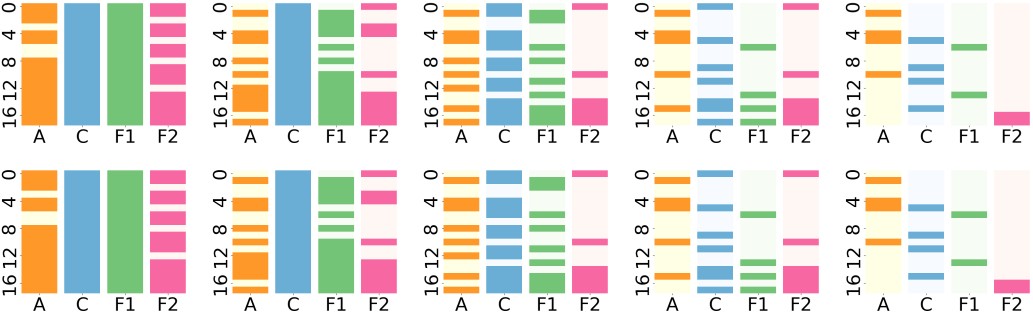

Figure 44: Visualization of $\mathbb{E}[g_{\text{enc}}]$ when encoder was pruned separately. The columns represents the $s_{\text{target}}$, with $s_{\text{target}}$ being 0.1, 0.3, 0.5, 0.7, and 0.9 from left to right. The first row represents the Italian-to-French, the second row represents Italian-to-German translation. The other settings are consistent with those in Figure 3.

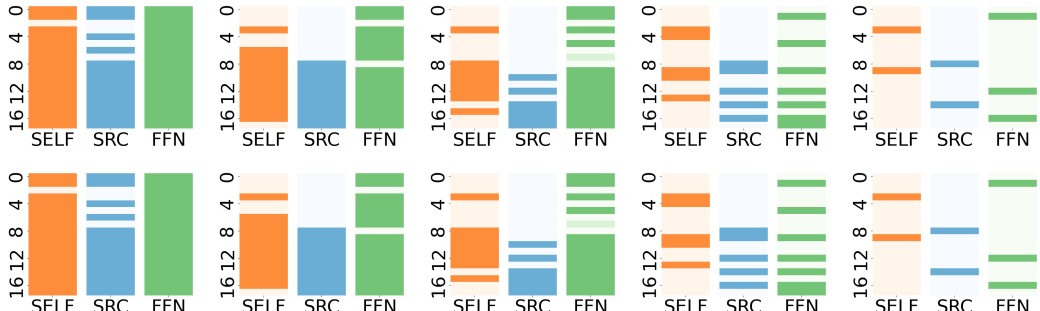

Figure 45: Visualization of $\mathbb{E}[g_{\text{dec}}]$ when decoder was pruned separately. The columns represents the $s_{\text{target}}$, with $s_{\text{target}}$ being 0.1, 0.3, 0.5, 0.7, and 0.9 from left to right. The first row represents the Italian-to-French, the second row represents Italian-to-German translation. The other settings are consistent with those in Figure 3.

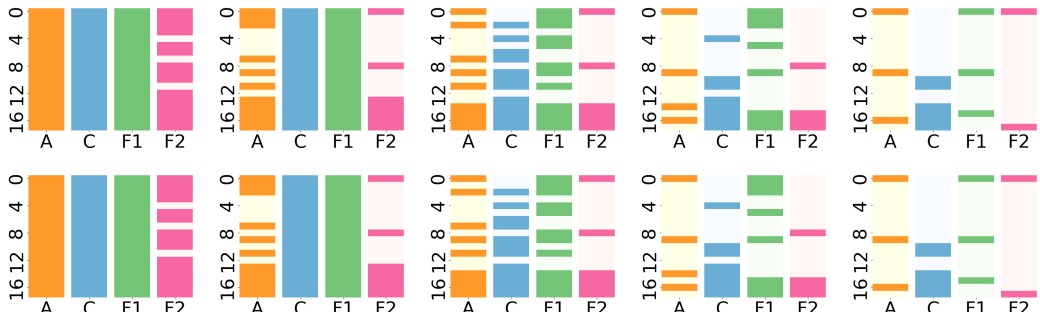

Figure 46: Visualization of $\mathbb{E}[g_{\text{enc}}]$ when encoder was jointly pruned with decoder. The columns represents the $s_{\text{target}}$, with $s_{\text{target}}$ being 0.1, 0.3, 0.5, 0.7, and 0.9 from left to right. The first row represents the Italian-to-French, the second row represents Italian-to-German translation. The other settings are consistent with those in Figure 3.

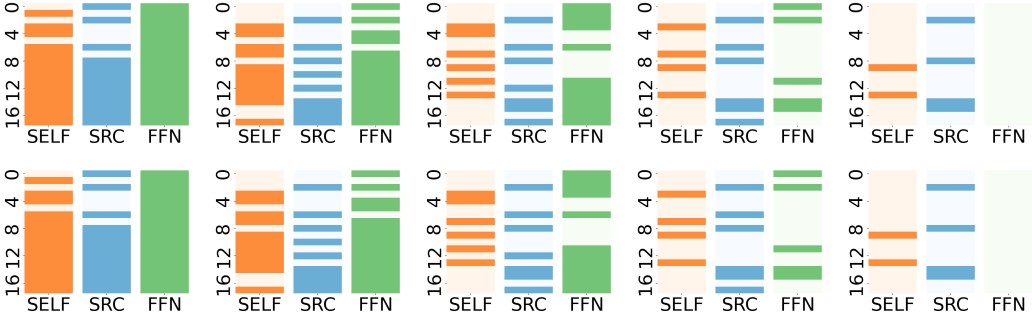

Figure 47: Visualization of $\mathbb{E}[g_{\text{dec}}]$ when decoder was jointly pruned with encoder. The columns represents the $s_{\text{target}}$, with $s_{\text{target}}$ being 0.1, 0.3, 0.5, 0.7, and 0.9 from left to right. The first row represents the Italian-to-French, the second row represents Italian-to-German translation. The other settings are consistent with those in Figure 3.

