# OpenReview forum: "Context-aware Dynamic Pruning for Speech Foundation Models"
_ICLR.cc/2025/Conference — ICLR 2025 Poster_

### Official Review · Reviewer_hGS7 · 2024-11-01

**Soundness:** 3
**Presentation:** 4
**Contribution:** 4
**Rating:** 8
**Confidence:** 4

**Summary:**

This paper addresses the problem of structured pruning for speech foundation models in the presence of context. The structured pruning is dynamically adjusted based on context during inference.

For each structured component in the model, a module selector estimates a mask, framed as a two-class classification problem, using Gumbel softmax. Experimental results on Speech Transcription and Speech Translation demonstrate the proposed approach’s effectiveness in improving inference efficiency without compromising performance.

The structured pruning also reveals key characteristics of several modules in the OWSM 3.1 architecture. Notably, pruning a decoder yields a significant speed-up compared to pruning an encoder.

**Strengths:**

- The paper underscores the effectiveness of the proposed structured pruning method with robust experimental results.

- The ablation studies are thoughtfully designed, and each section provides detailed insights into the impact of various components in encoder, decoder, and joint pruning.

- This work may inspire further research into new model architectures.

**Weaknesses:**

The original OWSM 3.1 model is a multilingual ASR model that covers a wide range of languages, each represented within the model's parameters. For high-resource languages, such as the one chosen in this study, pruning a significant portion of the network is often feasible without substantial performance degradation. However, for low-resource languages, the effects of similar pruning may lead to severe performance deterioration. Analyzing these effects could help clarify the limitations of this approach.

**Questions:**

1. In Table 1, the inference time for the encoder-decoder model with s_target = 0.1 exceeds the baseline (10.39 vs. 9.28). Does this indicate that model selection (pruning) is computationally expensive?

2. How critical is Gumbel-softmax for dynamic pruning? Could a simpler sigmoid-based selection function perform comparably in this setting? How are the temperatures for Gumbel-softmax adjusted?

3. Based on this study, can we reasonably infer that the E-Branchformer architecture contains additional components that could be pruned for efficiency, potentially aligning it more closely with the Conformer architecture?

---

> ### Author Response · Authors · 2024-11-26
>
> Thank you for your encouraging response.
>
> > Concern about the low-resource languages
>
> Thank you for pointing this out.
> To address this, we reviewed the pretraining data used by OWSM-V3.1 to confirm the data volume for each language.
> Based on this analysis, we incorporated Hungarian and Bulgarian into Europarl-ST for additional training.
> The data for these two languages was sourced from the Fleurs dataset and combined with Europarl-ST.
> The data volume used for OWSM pretraining for each language is as follows:
>
> | Language   | Data Size (h) |
> |------------|---------------|
> | French     | 2489          |
> | German     | 3704          |
> | Italian    | 707           |
> | Hungarian  | 97            |
> | Bulgarian  | 18            |
>
> Below, we present the evaluation results for decoders and encoders pruned individually. A sparsity of 0% represents the baseline model, which is simply fine-tuned without pruning.
>
> **Encoders**
> | Target Sparsity | Hungarian | Bulgarian |
> |-----------------|-----------|-----------|
> | 0.0             | 33.3      | 23.9      |
> | 0.1             | 38.7      | 38.2      |
> | 0.3             | 42.5      | 37.7      |
> | 0.5             | 42.1      | 38.3      |
> | 0.7             | 54.4      | 44.2      |
> | 0.9             | 89.2      | 78.8      |
>
> **Decoders**
> | Target Sparsity | Hungarian | Bulgarian |
> |-----------------|-----------|-----------|
> | 0.1             | 35.1      | 29.5      |
> | 0.3             | 34.1      | 24.2      |
> | 0.5             | 40.1      | 29.1      |
> | 0.7             | 36.3      | 26.5      |
> | 0.9             | 42.1      | 36.2      |
>
>
> Pruning the decoder led to declines in accuracy, similar to what was observed for higher-resource languages such as French and German. However, the impact of pruning the encoder appears to be more significant for low-resource languages. Even at lower sparsity ratios, encoder pruning caused a greater degradation in performance.
>
> This highlights the critical role of the encoder in preserving accuracy for low-resource languages. Since these languages typically have less training data, the encoder's ability to effectively capture the input feature is essential. In contrast, the decoder appears less sensitive to sparsity changes, likely because its task relies on the representations from the encoder.
>
> We have added this point to the Appendix B.
>
> > How critical is Gumbel-softmax for dynamic pruning? Could a simpler  sigmoid-based selection function perform comparably in this setting? How are the temperatures for Gumbel-softmax adjusted?
>
> Attempting to fine-tune OWSM-v3.1 with a sigmoid-based function is unlikely to work effectively. Through various experiments with different hyperparameter settings, we found that while training may succeed when the gate predictor outputs continuous values, issues often arise during inference.
> Specifically, even when the softmax temperature was lowered to make the gate predictor outputs more sharply peaked, we observed that the gate probabilities frequently converged to values around 0.4–0.6. This means that during training, some module outputs were scaled by 0.4, but during inference, those same modules were completely skipped. This mismatch caused a significant drop in inference accuracy.
>
> The Gumbel-Softmax approach we adopted addresses this issue by leveraging the Straight-through Gumbel-Softmax Estimator (SGSE), which enables the use of strictly binary values as pruning masks. With SGSE, the pruning mask is fully binary (0 or 1) during the forward pass in training, ensuring consistency between training and inference and maintaining robust performance.
> We recognize that this critical detail was not adequately explained in the paper. To address this, we have added a description in Section 3.3.
>
> > Based on this study, can we reasonably infer that the E-Branchformer  architecture contains additional components that could be pruned for  efficiency, potentially aligning it more closely with the Conformer  architecture?
>
> While there are similarities between the pruned E-Branchformer and Conformer architectures, the parallel structure of the E-Branchformer appears to be beneficial in the later layers, where modules remain fully active even at sparsity levels between 50% and 70%.
>
> In models with a large number of stacked E-Branchformer layers, our results suggest the possibility of reducing the reliance on self-attention in the earlier layers. While the computational cost of Conformer and E-Branchformer layers is comparable[1], replacing some of the earlier layers with convolution-based layers that place less emphasis on self-attention could be an approach to improve efficiency.
>
>
> **Reference**
> [1] Peng, Y., Kim, K., Wu, F., Yan, B., Arora, S., Chen, W., Tang, J., Shon, S., Sridhar, P., Watanabe, S. (2023) A Comparative Study on E-Branchformer vs Conformer in Speech Recognition, Translation, and Understanding Tasks. Proc. INTERSPEECH 2023, 2208-2212, doi: 10.21437/Interspeech.2023-1194

---

### Official Review · Reviewer_Eam6 · 2024-11-02

**Soundness:** 2
**Presentation:** 2
**Contribution:** 2
**Rating:** 6
**Confidence:** 4

**Summary:**

The paper proposes a method to prune (specifically structured pruning) Whisper-like speech foundation models. The authors first show the dependency of optimal pruned networks on several auxiliary characteristics of the task, languages, etc. and finally show that their method can achieve similar performance on benchmark datasets with similar performance.

**Strengths:**

- The paper is well written and is easy to read.
- The paper targets an important task of efficiency in speech models. As speech models continue to grow, efficiency is important and is an important research area.
- The proposed modification to the Context-Aware Gate Predictor is novel. To handle multiple languages and tasks simultaneously, the authors created vectors representing the language and task, combined them with the speech features, and used them as input to the Gate Predictors.
- The analysis provided on sparse encoders and decoders is novel. The extension to multilingiuality and multi-task is new and promising.

**Weaknesses:**

- A major weakness of the paper is the lack of baselines and comparison with prior-art. While I understand that the paper proposes a structured pruning mechanism, the ASR community has built several unstructured pruning mechanisms [1,2,3]. Unstructured pruning methods have a motivation similar to this paper and [1] also show that they achieve similar performance with high amounts of pruning. These methods have not been discussed or compared neither used as baselines.
- The paper feels like an analysis paper. Most methods applied are borrowed from Peng et al. (2023b), who also propose module-wise structured pruning. This paper just applies it to multilingual and multi-task scenarios and explores how a large-scale speech foundation
model adapts its structure based on context.
- The scope of the experimental section is rather narrow. The authors just show results on the Europarl-ST dataset and OWSM. A broader scope with more datasets and models would have been appreciated. Can Whisper be used? Can the authors expand the datasets employed for evaluation?
- (Minor) I am not amazed by the results. It  is well known that these networks are sparse. This paper proposes a method to prune these networks. In my experience, pruning is not done extensively as it might lead the model to loose performance in other areas beyond benchmark datasets. For example, given every layer stores different kinds of information, pruning a layer based on ASR results might lead useful information for another language/task to deteriorate. I understand this is an academic paper and nevertheless the performance shown in this paper are somewhat promising. This point will not affect my scores.

### Citations
[1] Lai, Cheng-I. Jeff, et al. "Parp: Prune, adjust and re-prune for self-supervised speech recognition." Advances in Neural Information Processing Systems 34 (2021): 21256-21272.
[2] Lodagala, Vasista Sai, Sreyan Ghosh, and Srinivasan Umesh. "Pada: Pruning assisted domain adaptation for self-supervised speech representations." 2022 IEEE Spoken Language Technology Workshop (SLT). IEEE, 2023.
[3] Fu, Yonggan, et al. "S $^ 6$-DAMON: Unlocking Structured Sparsity in Self-Supervised Speech Models via Data-Model Co-Compression."

**Questions:**

- Is elapsed time the correct metric for comparing model efficiency? In my experience, any torch implementation does not yet improve efficiency in terms of flops as torch is not yet designed for it. Can the authors show metrics in terms of FLOPs maybe?
- Please add wha conclusion we should draw from a figure to the caption of each figure. This makes the figures difficult to follow.

 don't have much questions. The findings presented in the paper are sound. However, I do not find anything much novel about the paper. Though novelty is not always that is required, but specifically and extension of an existing method to multilingual and multi-task scenarios is not very interesting to me. I would appreciate if the authors can point out whats novel in their "novel context-aware pruning technique" (except the gating technique) which is difficult for me to understand.

---

> ### Author Response · Authors · 2024-11-26
> **Response to reviewer Eam6 (1/2)**
>
> Thank you very much for your thoughtful and insightful comments.
>
> > A major weakness of the paper is the lack of baselines and comparison  with prior-art.
>
> We sincerely appreciate this insightful comment. To address this, we have incorporated new results using unstructured pruning methods, specifically magnitude-based pruning and PARP[1]. The table below presents a comparison between these two approaches. Fine-tuning for PARP was performed on a training dataset that included both ASR and ST, and WER was compared on the German ASR task.
>
> | Model                                      | WER  |
> |--------------------------------------------|------|
> | Original model (without fine-tuning)       | 24.6 |
> | + Unstructured pruning (sparsity = 0.1)    | 24.5 |
> | + Unstructured pruning (sparsity = 0.3)    | 24.5 |
> | + PARP (sparsity = 0.1, applied to encoder)| 31.2 |
> | + PARP (sparsity = 0.3, applied to encoder)| 31.8 |
> | + PARP (sparsity = 0.1, applied to decoder)| 19.8 |
> | + PARP (sparsity = 0.3, applied to decoder)| 16.4 |
>
> Even with unstructured pruning, it was evident that pruning the decoder keeps its performance. However, compared to module-level results at the same sparsity ratio, module-level pruning, especially on the encoder side, maintained accuracy more effectively.
> We mentioned this additional experiments in section 2 and have been added to the Appendix D.
> We are currently doing our best to tune the parameters for both encoder and decoder side. If we are able to train the model with better result, we will update the table and parameter settings in the Appendix.
>
> > The paper feels like an analysis paper.
>
> We fully acknowledge that our work builds upon the Peng et al. (2023). However, our approach introduces significant differences, particularly in the depth of analysis conducted, with a specific focus on module-wise pruning strategies. Furthermore, we have made key adjustments that allow the model to be trained to achieve a targeted sparsity level. While Peng et al. (2023) confined their analysis to sparsity levels ranging from approximately 60–90%, our work extends this analysis to include efficient module selection strategies across a broader range of sparsity levels, including lower sparsity settings.
> Additionally, we have modified the use of Gumbel-Softmax in our approach. Instead of employing the standard Gumbel-Softmax, we utilized the Straight-through Gumbel-Softmax Estimator (SGSE) to enable training with strictly binary pruning masks. This important distinction was not adequately detailed in the initial version of our paper; therefore, we have added a comprehensive explanation in Section 3.3 to clarify this point.
>
> > The scope of the experimental section is rather narrow.
>
> Similar concerns regarding the dataset were also raised by other reviewers, and we have conducted additional experiments incorporating VoxForge, Fleurs, and LibriSpeech from different perspectives.
> Regarding the suggestion to expand the scope of model analysis, as stated in Section 3.2, we determined that OWSM is more suitable for these experiments compared to Whisper.
>
>
> **Reference**
> [1] Lai, Cheng-I. Jeff, et al. "Parp: Prune, adjust and re-prune for self-supervised speech recognition." Advances in Neural Information Processing Systems 34 (2021): 21256-21272.

---

> ### Author Response · Authors · 2024-11-26
> **Response to Eam6 (2/2)**
>
> > Is elapsed time the correct metric for comparing model efficiency? In my experience, any torch implementation does not yet improve efficiency in terms of flops as torch is not yet designed for it. Can the authors  show metrics in terms of FLOPs maybe?
>
> We believe that elapsed time provides a practical perspective for comparing model efficiency in real-world scenarios, as it captures hardware and implementation-specific factors. Additionally, we appreciate your suggestion concerning complexity, and we included FLOPs for theoretical efficiency in table 1.
>
> The FLOPs for the encoder, decoder, and encoder-decoder at various sparsity levels are shown below. These values, measured in GFLOPs, were calculated using the `fvcore` package. The table presents the target sparsity ratios in descending order: 0.1, 0.3, 0.5, 0.7, and 0.9, with the non-sparse baseline having a value of 3781. The significant reduction in FLOPs for the decoder is attributed to its autoregressive inference process, where the beam size is treated as the batch size. In this analysis, we set the beam size to 5, meaning the encoder processes a batch size of 1 per inference, while the decoder processes a batch size of 5. These results have been incorporated into Section 4.
>
> | Encoder | Decoder | Enc-Dec |
> |---------|---------|---------|
> | 3697    | 3268    | 2843    |
> | 3690    | 2293    | 2249    |
> | 3669    | 1713    | 1698    |
> | 3633    | 1272    | 1139    |
> | 3613    | 625     | 682     |
>
>
> > Please add what conclusion we should draw from a figure to the caption of each figure. This makes the figures difficult to follow.
>
> Thank you for your valuable feedback.
> The figures in our paper can be broadly categorized into two types: those illustrating pruning patterns and those evaluating performance metrics.
> For performance evaluation graphs, we agree with your observation and have updated the captions to include key conclusions drawn from the figures.
> As for the figures related to pruning patterns, they are primarily used for comparative analysis across multiple figures.
> The detailed discussions and insights derived from these comparisons are provided in the main text.
>
>
> >  I would appreciate if the authors can point out whats novel in their "novel context-aware pruning technique" (except the gating technique) which is difficult for me to understand.
>
> We apologize for not clearly stating the differences in our use of Gumbel-Softmax during training. We employed the Straight-Through Gumbel-Softmax Estimator, which ensures that the computational graphs are fully aligned between training and inference.
> We have added detailed descriptions in Section 3.3.

---

> > ### Comment · Reviewer_Eam6 · 2024-11-27
> > **Thank You for your response**
> >
> > Thank You for providing responses to my questions. I would still request to:
> >
> > > Please add what conclusion we should draw from a figure to the caption of each figure.
> >
> > This is essential for readability to mention the key takeaway. I look forward to this.
> >
> > I am raising my score to 6 as my concerns are addressed and the paper is sound after revisions. However, I am not very impressed by the novelty of the approach as compared to Peng et al.
> >
> > Thank You again.

---

> > > ### Author Response · Authors · 2024-11-30
> > >
> > > Thank you very much for your time and positive feedback!
> > >
> > > >> Please add what conclusion we should draw from a figure to the caption of each figure.
> > > >
> > > > This is essential for readability to mention the key takeaway. I look forward to this.
> > >
> > > We believe that the figures illustrating pruning patterns may not have definitive conclusions on their own, some do exhibit interesting characteristics. We will include comments highlighting these points in the captions.

---

### Official Review · Reviewer_AdTV · 2024-11-03

**Soundness:** 3
**Presentation:** 4
**Contribution:** 3
**Rating:** 8
**Confidence:** 4

**Summary:**

The work proposes a dynamic pruning method to make large speech models more efficient during inference. Instead of using a fixed pruning structure, the authors adaptively prune parts of the model based on the specific context—such as the language, task, and speaker characteristics—while keeping the model’s core structure intact. This approach aims to reduce inference computational costs without losing accuracy, especially in multilingual and multi-task scenarios. Experimental results show that the proposed method achieves up to 35% faster inference with minimal impact on accuracy for tasks like automatic speech recognition (ASR) and speech translation (ST). Key contributions include the development of a context-aware pruning mechanism and a detailed analysis of how different tasks and languages benefit from tailored pruning strategies.

**Strengths:**

1) The approach is novel, introducing a dynamic pruning method with learned masking that adapts based on language and task contexts. The use of language tags to inform mask generation is particularly innovative, with strong potential for real-world impact.

2) The method is technically well-executed, with careful design of the module-level pruning strategy and gate predictor. Experiments cover multiple tasks (ASR and ST), languages, and configurations (encoder-only, decoder-only, and joint pruning), demonstrating the method’s generalizability across varied settings. The authors include detailed subtask-specific performance analyses, which reassures method does not introduce potential bias. Also authors show particular attention to avoiding data contamination—a notable strength given the field’s reproducibility challenges.

3) This work is promising for deploying large speech models in resource-limited environments, reducing inference time without sacrificing much accuracy. The context-aware pruning method offers an efficient, adaptable solution that could inspire further research into resource-saving models in NLP, computer vision, and other domains.

**Weaknesses:**

1) **Limited Baseline Comparisons:** The paper lacks comparisons with other dynamic pruning methods, particularly layer-wise pruning. The claim that finer-granularity pruning, like kernel or layer pruning, would disrupt model adaptability (lines 149-152) could be misleading, as studies such as [1] show that layer-wise pruning can maintain or even enhance model performance. Providing evidence for this claim, for instance by adding layer-wise pruning as a baseline, would provide valuable context for assessing the benefits of context-aware pruning, especially since there is a risk the current model is overly complex, potentially reducing its efficiency.
2) **Reproducibility:** The paper is clear enough to support basic reproduction. However, providing the code would greatly benefit the community and improve reproducibility. Additionally, a brief explanation and citation for Gumbel-Softmax [4] would enhance clarity, as this technique, though widely known, is central to the gate predictor setup.
3) **Need for more In-Depth Ablation Discussion:** While the paper analyzes different pruning configurations (encoder-only, decoder-only, joint pruning) with heatmaps for various subtasks being provided in the Appendix, it lacks discussion on how different languages or tasks are affected by these setups. Some languages may benefit more or less from the pruning process, potentially relating to their resourcefulness levels in OWSM pre-training data. Such insights could clarify the impact of context-aware pruning across subtasks.
4) **Interpretability of Pruning Choices:** The paper discusses the interpretability of context-based pruning but could offer deeper insights. For example, quantitative metrics or visualizations on why certain modules are pruned would help clarify the gate predictor’s influence on pruning decisions and whether these choices are consistent across contexts. For example, you could study the correlation between pruning decisions and specific linguistic features of the input. Additionally, analyzing the impact of pruning only specific parts of the layers (e.g., self-attention vs. FFN) could improve understanding, as prior studies [2, 3] suggest these components may have varying importance across tasks and have different weight on the computational costs.
5) **Contribution Clarification:** Network pruning using learned masks is not particularly novel, and have been studied in various domains, including self-supervised models for language and speech. Expanding on these methods in the related work section and clarifying the novelty of this specific approach would strengthen the paper’s positioning.

### References
[1] Raposo, D., Ritter, S., Richards, B., Lillicrap, T., Humphreys, P. C., & Santoro, A. (2024). Mixture-of-Depths: Dynamically allocating compute in transformer-based language models.


[2] Zhang, B., Bapna, A., Sennrich, R., & Firat, O. (2021). Share or not? Learning to schedule language-specific capacity for multilingual translation. ICLR 2021.


[3] Ferraz, T. P., Boito, M. Z., Brun, C., & Nikoulina, V. (2024). Multilingual Distilwhisper: Efficient Distillation of Multi-Task Speech Models Via Language-Specific Experts. ICASSP 2024.


[4] Jang, E., Gu, S., & Poole, B. (2017). Categorical Reparameterization with gumbel-softmax. ICLR 2017.

**Questions:**

- How significant are the results? Can you provide information on the statistical significance of your results, such as confidence intervals? Reporting confidence intervals or statistical tests would strengthen the findings, especially since many works in adaptation and fine-tuning report these metrics to assess consistency and robustness.
- Have you observed any patterns in performance across different subtasks or languages, particularly with varying sparsity targets?
- Does your method require specifying the source language at inference, or can the model identify the language automatically, as Whisper does?
- Would it be possible to provide code or configuration files to facilitate reproducibility?

As a side note, I recommend adding baseline (without pruning) to the tables as this would help visualization and understanding.

---

> ### Author Response · Authors · 2024-11-25
> **Response to reviewer AdTV (1/2)**
>
> Thank you for providing such a valuable review.
>
> > Limited Baseline Comparisons
>
> Based on reviewer's feedback, we conducted additional experiments involving layer skipping, reproduction of , and unstructured pruning.
> However, considering the objective of our study—to analyze and address the research question—we would like to emphasize that skipping layers reduces the granularity of analysis compared to module-based pruning. This distinction is important for ensuring clarity in our findings.
>
>
> > Reproducibility:
>
> We conducted our experiments with ESPnet, and uploaded our experimental codes to anonymized github.
> https://anonymous.4open.science/r/anonymized_dynamic_pruning-C546/
>
> We have also added the citation for Gumbel-Softmax, and added more detailed explanation on how we utilized the Straight-through Gumbel Softmax Estimator. Thank you very much for bringing this to our attention.
>
> > Need for more In-Depth Ablation Discussion:
>
> Thank you for your observation regarding the need for a more in-depth ablation discussion based on pre-training dataset of OWSM model.
> Since all the pretraining data is available, we checked the data distribution according to the paper, specifically for French, German, and Italian. The data sizes used for OWSM pre-training is as follows:
>
> | Language | Data Size (h) |
> |----------|---------------|
> | French   | 2489          |
> | German   | 3704          |
> | Italian  | 707           |
>
> Given the smaller amount of data for Italian, we conducted additional analyses to investigate whether there were significant differences compared to other languages or tasks. However, we did not identify any noteworthy findings.
> In response, we are also examining languages with even smaller pretraining datasets, such as Hungarian and Bulgarian, which contain 97 hours and 18 hours of data, respectively. We incorporated these analyses and findings into the revised manuscript, specifically in Appendix B.
>
> > Interpretability of Pruning Choices:
>
> We deeply appreciate your insightful comments. To address your feedback, we included additional discussions based on the characteristics of each tasks on the following points:
>
> **Joint Pruning in Section 4.2.1**
> - When only the decoder is pruned, early src-ATT layers are removed. In contrast, when both the encoder and decoder are pruned together, earlier src-ATT layers in the decoder become more active.
> - This difference likely stems from whether all encoder modules are fully utilized. If encoder capacity is limited, the decoder compensates by computing self-ATT and FFN layers on the audio features. If the encoder is fully available, the decoder focuses on incorporating more contextual information from output tokens before src-ATT.
>
> **Differences Between ASR and ST in Figure 7**
> - Self-ATT is more prominent before src-ATT in ST than in ASR.
> - This is likely because alignment via attention is more critical in ST, whereas ASR prioritizes directly integrating audio features, leading to differences in the timing of self-ATT computations.
> We will incorporate these discussions into the manuscript, including a detailed analysis in Section 4.2.1 and 4.2.2.
>
> > Contribution Clarification:
>
> In this work, we utilized the Straight-through Gumbel-Softmax Estimator (SGSE) to create a trainable pruning mask that outputs strictly binary values for training. We regret that this important detail was not clearly articulated in the paper.
>
> For example, during training, the gate probabilities would be multiplied to the output of each modules , and during inference, activation is determined using a threshold value ranges from 0 to 1.
> After training our model, we realized that there are several values close to the threshold.
> This means that use partially activate those modules during training, while we completely ignore these modules during inference.
> We solved this issue by using the SGSE.
>
> We have added a paragraph to Section 3.3 to explain this approach and included a detailed problem formulation in the Appendix A.
>
>
> > Have you observed any patterns in performance across different subtasks  or languages, particularly with varying sparsity targets?
>
> We understand this comment as relating to the need for an in-depth analysis, as previously noted. For further details, please refer to that section.

---

> ### Author Response · Authors · 2024-11-25
> **Response to reviewer AdTV (2/2)**
>
> > Does your method require specifying the source language at inference, or can the model identify the language automatically, as Whisper does?
>
> Our method requires specifying the source language at inference time; it does not incorporate an automatic language identification process as Whisper does. This is because our objective is to analyze how a pre-trained foundation model manages task-specific and language-specific differences. We believe that incorporating language ID estimation is not necessary to our work.
>
>
> > Would it be possible to provide code or configuration files to facilitate reproducibility?
>
> We have uploaded our experimental code to an anonymous github. Please refer to the following link: https://anonymous.4open.science/r/anonymized_dynamic_pruning-C546/
>
> > As a side note, I recommend adding baseline (without pruning) to the tables as this would help visualization and understanding.
>
> Thank you for your suggestion. While we have not yet included the baseline (without pruning) in the current tables due to the large number of models being compared, we explor ways to adjust the table layout to incorporate this information more effectively. We will make our best effort to present this in a concise manner in the revised version.

---

> > ### Comment · Reviewer_AdTV · 2024-11-26
> > **Response to Part 2 of Authors rebuttal**
> >
> > Thank you for providing the code, and clarifying about the language definition. I will wait the pdf update to discuss the part 1.

---

> > > ### Author Response · Authors · 2024-11-26
> > >
> > > Thank you for your prompt response.
> > >
> > > We'd like to let you know that we’ve updated the PDF.

---

> > > > ### Comment · Reviewer_AdTV · 2024-11-28
> > > > **Response to part 1 of Authors rebuttal**
> > > >
> > > > Thanks for have updated the PDF and for the comments.
> > > >
> > > > I appreciate the extra experiments done, and I would highlight the interesting limitation of the method regarding low-resource languages. Thanks again for you efforts.
> > > >
> > > > Some final recommendations:
> > > > - I recommend point to the exactly code in which the work contributed to in the ESPnet library.
> > > > - Clarify the term "LayerSkip" considering the existence of homonym method [1].
> > > >
> > > > To conclude, I believe my main concerns were satisfied and I also took some time to read the concerns of other reviewers, which I also think are well addressed, so I raised my score accordingly.
> > > >
> > > > [1] Elhoushi, Mostafa, et al. "Layer skip: Enabling early exit inference and self-speculative decoding." arXiv preprint arXiv:2404.16710 (2024).

---

> > > > > ### Author Response · Authors · 2024-11-30
> > > > >
> > > > > Thank you very much for your valuable feedback and positive response! We’re pleased that your concerns have been addressed. Below, we’ve addressed each of your comments in detail:
> > > > >
> > > > > > I recommend point to the exactly code in which the work contributed to in the ESPnet library.
> > > > >
> > > > > Our work primarily involved the following files in the ESPnet library:
> > > > > - espnet2/asr/encoder/e_branchformer_ddn_token_condition_encoder.py (for pruning the encoder)
> > > > > - espnet2/asr/decoder/transformer_ddn_token_condition_decoder.py (for pruning the decoder)
> > > > >
> > > > > While we also made modifications to other scripts to incorporate these changes into the training and evaluation processes, the two files above are the key contributions of our work.
> > > > >
> > > > > > Clarify the term "LayerSkip" considering the existence of homonym method [1].
> > > > >
> > > > > Thank you for pointing this out. We will add a sentence to clarify the term.

---

> ### Comment · Reviewer_AdTV · 2024-11-26
> **PDF is not available**
>
> I believe the pdf was not updated (at least for me it seems to be the same thing). Can you check that? I couldn't verify the results you added.
>
> Also the authors did not address the question about significance of the results. Several conclusions were made on pretty narrow differences, so I wonder if authors can provide more information about it.

---

> > ### Author Response · Authors · 2024-11-30
> >
> > > Also the authors did not address the question about significance of the results
> >
> > Thank you for highlighting this point, and we apologize for the delay in addressing it. Could you kindly clarify which specific results you are interested in regarding statistical significance?

---

### Official Review · Reviewer_pWFs · 2024-11-04

**Soundness:** 3
**Presentation:** 3
**Contribution:** 2
**Rating:** 6
**Confidence:** 4

**Summary:**

This research paper introduces a context-aware dynamic pruning technique for speech foundation models, enabling the models to adjust their structure based on contextual factors like language and task during inference. The authors showcase the effectiveness of their approach in a multilingual, multi-task setting, achieving approximately 30% reductions in inference time while maintaining BLEU score, ST task, and little degradation on the WER, ASR task. Additionally, they analyze the pruned model structure, offering insights into the varying importance of specific modules for different tasks and highlighting potential optimizations for speech processing architectures.

**Strengths:**

The paper's main strength lies in its analysis of the importance of different modules for various speech processing tasks. The findings indicate that, within the encoder-decoder framework, pruning the decoder is more effective for speech translation (ST) and automatic speech recognition (ASR) tasks. The paper is well-written and easy to understand with minor typos and incomplete sentences.

**Weaknesses:**

The size of dataset (20 hours) used is too small to make general claims.

The authors do not compare their results with any other pruning methods. Like the effect of simply removing the layers, instead of different modules within, on the performance on the given the size of the dataset used in the study.

While the paper focuses on reducing inference time, it doesn't address the computational overhead introduced by the dynamic pruning process itself and searching the best sparsity number during finetuning.

Lack major novelty.

**Questions:**

Typo ? [line 130] if out validation data >> if validation data

I think [at line 285] the sentence is incomplete.

At line 285, the authors make a statement "This supports our initial question regarding the need for a large-scale decoder in ASR systems". This is different form the results in the paper, pruning the decoder a lot does not effect ASR performance much compared to encoder as shown in Figure 2.

In Figure 1, it is not clear how mean pooling can be applied to the input of the transformer decoder given its autoregressive nature.

Not able to understand after reading the paper, if the time reduction is from using the joint finetuned model or individual task finetuned model. I think it is for individual task finetuned models.

[Please correct me if I am wrong]. The paper simply extends the work of Peng et al. (2023b) by utilizing the model structure of a speech foundation model to address multilingual and multi-task scenarios and therefore lacks major novelty. Please provide more detailed comparison of their approach to previous work and their novel contributions beyond Peng et al. (2023b).

Not suited for joint pruning? As per the results, it is better to prune decoder in multitask setting. But authors report for the ASR and ST tasks SRC-ATT and SELF-ATT are individually important in the decoder. Therefore, the motivation to prune the modules in the joint finetuning is diluted.

---

> ### Author Response · Authors · 2024-11-25
> **Response to reviewer pWFs (1/2)**
>
> Thank you very much for reviewing our paper and for providing positive feedback.
>
> > The size of the dataset (20 hours) used is too small to make general claims.
>
> We appreciate your observation. Indeed, the limited size of the dataset was a valid concern. To address this, we conducted additional experiments by incorporating the VoxForge dataset, which includes the same languages, into the original dataset. This resulted in a joint dataset for ASR task, whose sizes are as follows:
>
> | Language | Europarl-ST | + voxforge |
> |---|---|---|
> | French | 21.06 | 41.04 |
> | German | 17.71 | 63.40 |
> | Italian | 20.93 | 37.00 |
>
> It is true that the absolute increase in data size is not substantial. However, we believe that this amount of additional data is sufficient to evaluate the impact of data increase on the performance of our method. Moreover, as our approach relies on fine-tuning a pre-trained model, using a dataset of this scale ensures practical and reproducible results.
> The results of the models trained on this expanded dataset are presented below. The table highlights the performance of the German ASR model with the decoder side pruned, showing improvements in accuracy compared to models trained on Europarl-ST alone:
>
> | Target Sparsity | Europarl-ST | + Voxforge |
> |-----------------|-------------|----------|
> | 0.1             | 15.3        | 14.8     |
> | 0.3             | 15.9        | 15.1     |
> | 0.5             | 20.4        | 19.3     |
> | 0.7             | 17.2        | 17.5     |
> | 0.9             | 24.5        | 25.5     |
>
> The performance degradation across different sparsity levels is comparable between Europarl-ST and joint dataset. To reflect these results, we have updated Section 4 and Appendix B to include these additional experiments.
>
> > The authors do not compare their results with any other pruning methods.
>
> Thank you for your feedback. In response to the feedback, we conducted additional experiments comparing module-based pruning, layer-level skipping, and unstructured pruning. These experiments were carried out using our joint ASR and ST dataset.
> We report the WER for German ASR, where the encoders and decoders were pruned separately. The sparsity patterns used in these experiments are illustrated in Figure 13 in Appendix C.
>
> **Encoder**
> | Target Sparsity | Module Skip | Layer Skip |
> |-----------------|-------------|------------|
> | 0.1             | 14.5        | 31.3       |
> | 0.3             | 15.0        | 32.0       |
> | 0.5             | 16.8        | 34.2       |
> | 0.7             | 25.1        | 40.2       |
> | 0.9             | 47.3        | 108.6      |
>
> **Decoder**
> | Target Sparsity | Module Skip | Layer Skip |
> |-----------------|-------------|------------|
> | 0.1             | 15.8        | 15.4       |
> | 0.3             | 16.4        | 15.2       |
> | 0.5             | 16.2        | 16.5       |
> | 0.7             | 18.3        | 18.7       |
> | 0.9             | 26.4        | 99.9       |
>
> The degradation in accuracy on the encoder side is significant, leading us to believe that layer-level pruning for encoder side is not well-suited for speech foundation models. With layer-level pruning, even cgMLP, which is preserved in module-level pruning, is forcibly removed. This results in the loss of parameters that play a critical role on the encoder side, highlighting a key disadvantage of this approach.
>
> For the decoder side, the results indicate that up to 70% sparsity, there is no significant difference between skipping at the layer level and skipping at the module level. However, at lower sparsity levels, the layer-level approach appears to perform better, whereas at higher sparsity levels, the module-level approach demonstrates superior performance. Based on these findings, we believe that at higher sparsity levels, the roles and processing order of individual modules are more effectively optimized compared to the more coarse-grained approach of skipping entire layers, leading to the observed results.

---

> ### Author Response · Authors · 2024-11-25
> **Response to reviewer pWFs (2/2)**
>
> We also calculated the FLOPs for these layer-skipped models. Since different modules are utilized depending on whether module-level or layer-level pruning is applied, we examined the resulting differences. The table below shows the FLOPs when the encoder is pruned. As a result, we found that module-level pruning results in slightly lower FLOPs compared to layer-level pruning. These findings further emphasize the importance of module-level pruning from a performance perspective. We will address these points in Section 3.2 and include additional details in the Appendix C.
>
> **GFLOPs**
>
> | Target Sparsity | Module Skip | Layer Skip |
> |-----------------|-------------|------------|
> | 0.1             | 3697        | 3697       |
> | 0.3             | 3690        | 3692       |
> | 0.5             | 3669        | 3666       |
> | 0.7             | 3633        | 3640       |
> | 0.9             | 3613        | 3620       |
>
> > While the paper focuses on reducing inference time, it doesn't address  the computational overhead introduced by the dynamic pruning process  itself and searching the best sparsity number during finetuning.
>
> Thank you. We calculated the FLOPs for each sparsity level and added them to Table 1.
> The computation overhead for determining sparsity during fine-tuning, including both the fine-tuning and evaluation processes, required approximately two days on a single A40 GPU.
>
> > Lack majour novelty
>
> Although our method is indeed an extension of Peng et al. (2023), we have introduced novel contributions compared to Peng et al. (2023) in the following aspects:
> - Multilingual and Multitask Training: We trained our model in a multilingual and multitask setting.
> - Different Utilization of Gumbel-Softmax: We employed Gumbel-Softmax differently in our approach: that is, we used the straight-through Gumbel-Softmax estimator for training pruning masks.
>
> We realized that we did not adequately describe the second point in our paper. To address this, we included additional explanations in the section 3.3 of the Methodology.
> Moreover, we noticed that our method lacks a detailed problem formulation. We have addressed this by pointing it out in section 3.3 and adding the formulation in the Appendix A.
>
> > Typo ? [line 130]
>
> Thank you for catching that! We have corrected it.
>
> > I think [at line 285] the sentence is incomplete.
>
> Thank you again for pointing this out. We have revised it to ensure the sentence is complete.
>
> > In Figure 1, it is not clear how mean pooling can be applied to the  input of the transformer decoder given its autoregressive nature.
>
> Thank you, the diagram could indeed be clearer. The mean pooling applies to the audio information, which is used to control the decoder. This does not interfere with the auto-regressive processing on the decoder side during inference.
> To clarify this further, we revised Figure 1 (a) and (b) to make it more apparent that this pertains to the audio information.
>
> > Not able to understand after reading the paper, if the time reduction is from using the joint finetuned model or individual task finetuned  model. I think it is for individual task finetuned models.
>
> We apologize for not making this clear in the paper. The model used for inference is trained separately for each task, with individual models fine-tuned for ASR and ST tasks. We included this information in the caption of table 1.
>
>
> >  Please provide more detailed comparison  of their approach to previous work and their novel contributions beyond  Peng et al. (2023b).
>
> We provided a detailed explanation from the perspective of lack of novelty in response to the previous comment. Specifically, we clarified the differences in the use of Gumbel-Softmax in Section 3.3 and added a detailed formulation in the Appendix A. We also reproduced their results for comparison and included these results in Appendix B.
>
>
> **Reference**
> 1. Yifan Peng, Jaesong Lee, and Shinji Watanabe. I3d: Transformer architectures with input-dependent
> dynamic depth for speech recognition. In ICASSP 2023 - 2023 IEEE International Conference
> on Acoustics, Speech and Signal Processing (ICASSP), pp. 1–5, 2023

---

### Meta-Review · Area_Chair_SiSp · 2024-12-22

**Metareview:**

> This research paper introduces a context-aware dynamic pruning technique for speech foundation models, enabling the models to adjust their structure based on contextual factors like language and task during inference. The authors showcase the effectiveness of their approach in a multilingual, multi-task setting, achieving approximately 30% reductions in inference time while maintaining BLEU score, ST task, and little degradation on the WER, ASR task. Additionally, they analyze the pruned model structure, offering insights into the varying importance of specific modules for different tasks and highlighting potential optimizations for speech processing architectures.

The reviewers all accepted this paper, which of sufficient scope and interest for ICLR.

**Additional Comments On Reviewer Discussion:**

The reviewers did not engage with the rebuttal, even though the authors did.

---

### Decision · Program_Chairs · 2025-01-22

Accept (Poster)